# Learning the histone codes with large genomic windows and three-dimensional chromatin interactions using transformer

Dohoon Lee ®[1,2], Jeewon Yang ®[3] & Sun Kim ®[3,4,5,6] ✉

The quantitative characterization of the transcriptional control by histone modifications has been challenged by many computational studies, but most of them only focus on narrow and linear genomic regions around promoters, leaving a room for improvement. We present Chromoformer, a transformer-based, three-dimensional chromatin conformation-aware deep learning architecture that achieves the state-of-the-art performance in the quantitative deciphering of the histone codes in gene regulation. The core essence of Chromoformer architecture lies in the three variants of attention operation, each specialized to model individual hierarchy of transcriptional regulation involving from core promoters to distal elements in contact with promoters through three-dimensional chromatin interactions. In-depth interpretation of Chromoformer reveals that it adaptively utilizes the long-range dependencies between histone modifications associated with transcription initiation and elongation. We also show that the quantitative kinetics of transcription factories and Polycomb group bodies can be captured by Chromoformer. Together, our study highlights the great advantage of attention-based deep modeling of complex interactions in epigenomes.

The control of gene expression is carried out by diverse groups of regulators, including transcription factors, coactivators, corepressors along with genomic sequence elements. However, the basic premise behind the interplay among these factors is the appropriate configuration of the covalent modifications of histone tails, or histone modifications (HMs), at the relevant genomic regions since they play a pivotal role in the regulation of chromatin accessibility. Thus, it can be conceived that an amount of HMs and their combinations encode the regulatory potential of the nearby genomic regions.

This notion is referred to as the 'histone code hypothesis'[1]. There have been a number of computational and quantitative approaches to crack the regulatory code of gene expression encoded by HMs. Most of them are predictive models that utilize the levels of HMs at promoters surrounding transcription start sites (TSSs) to predict the expression level of the corresponding gene. Notably, recent studies have shown the superior performance of deep learning models compared to the conventional machine learning models in this task[2,3].

To date, deep learning has been making remarkable breakthroughs in diverse fields of computational biology, ranging from the characterization of binding the specificity of DNA- and RNA- binding proteins[4] to the longstanding problem of the protein structure prediction based on its amino acid sequence[5]. These successes of deep learning in biology could not be achieved without the invention of novel model architectures and also their clever applications for complex biological problems. In that sense, the high complexity of histone code indeed made it a great target for deep learning, as shown in the existing approaches, but they still pose two major limitations that motivate the development of a new approach.

[1]Bioinformatics Institute, Seoul National University, Seoul 08826, Republic of Korea. [2]BK21 FOUR Intelligence Computing, Seoul National University, Seoul 08826, Republic of Korea. [3]Interdisciplinary Program in Artificial Intelligence, Seoul National University, Seoul 08826, Republic of Korea. [4]Interdisciplinary Program in Bioinformatics, Seoul National University, Seoul 08826, Republic of Korea. [5]Department of Computer Science and Engineering, Seoul National University, Seoul 08826, Republic of Korea. [6]AIGENDRUG Co., Ltd., Seoul 08826, Republic of Korea. ✉e-mail: sunkim.bioinfo@snu.ac.kr

One is that they could only use narrow genomic windows around TSSs. This is because the deep learning architectures that those models were based on, such as convolutional neural networks (CNNs) and recurrent neural networks (RNNs), were not effective in modeling the dependencies within long sequences. CNNs are highly specialized for learning local patterns of data, but it is challenging for them to learn the distant dependencies between the patterns. Although developed to model sequential data, RNN architectures also have difficulties in capturing the long-range dependencies clearly since the information embedded in a single position becomes gradually diluted and gets contaminated while the model computation travels along the positions between the two distant positions. Indeed, advanced forms of RNN cells such as Gated Recurrent Units[6] or Long Short-Term Memory (LSTM)[7] partially ameliorate this problem, but the intrinsic inefficiency in modeling long sequences due to the recurrence still remains.

Next, a majority of the deep learning models do not account for the distal *cis*-regulation mediated by three-dimensional (3D) chromatin folding, even though it has been widely known that the physical interactions between core promoters and distal *cis*-regulatory elements critically modulates the gene expression[8,9]. In other words, the regulatory information conveyed by the histone code is allowed to not only propagate locally, but also jump between distant genomic loci through 3D chromatin interactions[10]. Fortunately, the recent advancement of high-throughput measurement technologies such as Hi-C[11] succeeded in providing a fine-resolution view of 3D chromatin interactions at kilobase-scale and offered us with unprecedented opportunities for exploiting such valuable information to model the comprehensive view of gene regulation. There are few emerging studies that explicitly take the 3D chromatin interactions into consideration to predict the gene expression. One such example is GC-MERGE[12], a graph neural network (GNN) to propagate information between interacting genomic regions to predict the expression levels of genes. Although it is a proof-of-concept model that cannot be applied to genes without any chromatin interactions and only performs 10 kbp genomic bin-level predictions but not at gene-level, it still underscores the promise of modeling epigenomic contexts of distal genomic regions along with those of promoters.

Meanwhile, a deep learning model architecture named transformer, which was originally developed for natural language processing[13], has been exhibiting great potential for understanding the latent grammar of DNA sequences[14], amino acid sequences[15], and even their alignments[16]. In particular, in this study, we noticed that the two main functionalities of the transformer architecture are highly suitable to tackle the two aforementioned challenges. First, transformers can precisely model the long-range dependencies in sequential data. This is elegantly done by the addition of positional encodings to input sequences. These input features harboring positional information are treated independently and fed into a subsequent self-attention module which calculates the all-pairwise dependencies between the input features. Therefore, long-range dependencies can be captured without the interference of features located between the pair. Secondly, the transformer architecture can also be applied to model unordered sets of entities along with the interactions among them. Of note, this is not straightforward for most of the deep learning architectures since the operations comprising them depend on input positions. On the other hand, the operations comprising the transformer are basically permutation-invariant. The interactions between input features are only considered in self-attention operations, and all the other operations are done in a position-wise manner, so they can be applied to a model an unordered set of features. Together, these two strengths of the transformer architecture make it a promising choice for the quantitative modeling of histone codes by allowing us to utilize wider genomic windows near TSSs and histone codes at multiple distal regulatory regions simultaneously.

Here, we present a transformer-based deep learning architecture named Chromoformer to predict the gene expression levels based on the HMs at the wide neighborhood of TSSs as well as the HMs placed at the distal regulatory elements. Based on the model architecture consisting of three variants of self-attention operations to reflect the hierarchy of 3D gene regulation, Chromoformer achieves far better predictive power compared to the other deep learning models for gene expression prediction. Moreover, through the comprehensive investigation on how the use of transformer architecture contributed to the superior performance of the model, we demonstrate that the long-range modeling of epigenetic context near TSS and simultaneous integrative modeling of distal regulatory regions actually worked to improve performances. Finally, we show that we could draw artificial intelligence-driven hypotheses for the quantitative effect of *cis*-regulation by the two subdomains within nuclei, transcription factories, and silencing hubs, through the interpretation of the dynamics of latent embeddings of the regulatory states learned by Chromoformer.

## Results

### Chromoformer adopts three-level transformer architecture that reflects the hierarchy of 3D gene regulation

The core design principle of Chromoformer is twofold. One is to extract as much proximal regulatory information as possible from the HMs at the core promoters, and the other is to incorporate the distant histone codes whose information is transmitted to the core promoter through 3D chromatin interactions. To fully utilize the transformer architecture to model the complex dynamics of *cis*-regulations involving multiple layers, we conceptually decomposed the gene regulation into a three-layered hierarchy: (1) *cis*-regulation by core promoters, (2) 3D pairwise interaction between a core promoter and a putative *cis*-regulatory regions (pCREs) and (3) a collective regulatory effect imposed by the set of 3D pairwise interactions. To computationally emulate this hierarchy, we introduced three transformer-based submodules called Embedding, Pairwise Interaction, and Regulation transformers that are specialized to learn the respective grammar of gene expression regulation in the order of increasing complexity.

Before illustrating the model architecture, we briefly describe the input features used throughout this study. Chromoformer was trained using read depth values from histone ChIP-seq experiments for seven major HMs (H3K4me1, H3K4me3, H3K9me3, H3K27me3, H3K36me3, H3K27ac, and H3K9ac) (Supplementary Fig. 1a). Read depths were averaged and log2-transformed for fixed-sized bins across 40 kbp regions flanking TSSs (Fig. 1a and Supplementary Fig. 1b). To account for the distal *cis*-regulation, we additionally utilized the HM signals at pCREs that are known to interact with the core promoter in the corresponding cell type (Supplementary Fig. 1c). For that, an experimentally validated set of pCREs for each core promoter was obtained using a publicly available collection of promoter-capture Hi-C (pcHi-C) data[17]. The 3D chromatin interactions were characterized at the resolution of HindIII restriction fragments. The interactions were characterized at adequately high-resolution, as the median and average length of those fragments were 4797 bp and 5640 bp, respectively, and about 95% of them were less than 10 kbp (Supplementary Fig. 2).

The full model architecture used in this study is illustrated in Fig. 1b. At the highest level, it consists of three independent modules each of which accepts input features at different resolutions and in turn produces an embedding vector of the regulatory state at the core promoter. The resulting three regulatory embeddings are concatenated to form a multi-scale regulatory embedding which is subsequently fed into fully-connected layers to predict the expression level of the gene. The use of multi-scale regulatory embedding resulted in better performance than using any single-resolution regulatory embedding, and the combination of all three resolutions gave a robustly higher performance increase than the combination of any two resolutions (Supplementary Fig. 3). Meanwhile, combining

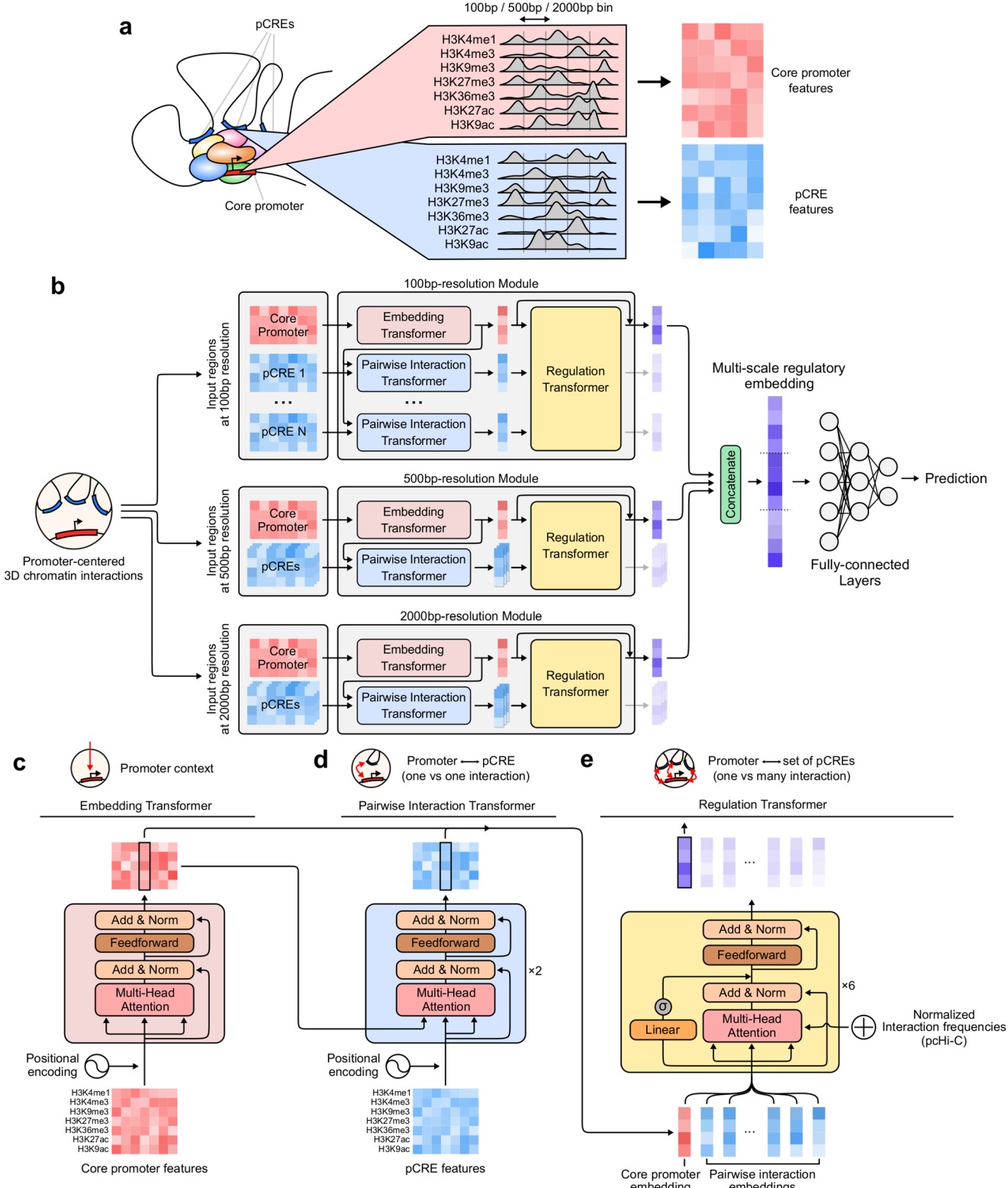

**Fig. 1 | Chromoformer model architecture. a** Input features. To predict the expression of a gene using levels of histone modifications (HMs), we extracted binned average signals of HMs from both the core promoter and putative *cis*-regulatory regions (pCREs). **b** Chromoformer architecture. Three independent modules were used to produce multi-scale representation of gene expression regulation. Each of the modules is fed with input HM features at different resolutions to produce an embedding vector reflecting the regulatory state of the core promoter. **c** Embedding transformer architecture. Position-encoded HM signals of core promoter features are transformed into core promoter embeddings through self-attention. **d** Pairwise Interaction transformer architecture. Position-encoded HM signals of pCREs are used to transform the core promoter embeddings into Pairwise Interaction embeddings through encoder-decoder attention. **e** Regulation transformer architecture. Using the whole set of the core promoter and Pairwise Interaction embeddings and gated self-attention, the Regulation transformer learns how the pCREs collectively regulate the core promoter. To guide the model to put greater attention to frequently occurring three-dimensional (3D) interactions, the normalized interaction frequency vector is added to self-attention affinity matrices.

regulatory embeddings at different resolutions using self-attention operation[13,18–20] did not seem to have significant advantages over concatenation, so we decided to concatenate regulatory embeddings for the sake of the simplicity of the model (Supplementary Fig. 4). The Embedding transformer (Fig. 1c) learns the histone codes acting at the direct vicinity of a TSS and produces a fixed-sized vector that summarizes the epigenetic state of the region. This submodule alone works very similar to the existing machine learning models for HM-based gene expression prediction, but we expected that the use of transformer architecture will allow the model to precisely identify relevant signals within a wide-range view (up to 40 kbp) of core promoter HM contexts without any performance degradation. Next, the resulting core promoter embeddings are further updated by the Pairwise Interaction transformer (Fig. 1d) in the context of pairwise *cis*-regulatory interactions between promoters and pCREs. Instead of using typical self-attention layers as in the Embedding transformer, this module is built with encoder-decoder attention layers. Since the activity of a promoter is modulated by the contact with pCREs, the encoder-decoder framework was chosen to reflect this by decoding the promoter embeddings given the context of the pCRE features. We call the resulting embedding vectors pairwise interaction embeddings as they carry the information of one-to-one relationship between promoters and pCREs. Finally, the Regulation transformer (Fig. 1e) accepts a union set of the core promoter and pairwise interaction embeddings and finally produces a regulatory embedding by integrating them. This module models the whole landscape of *cis*-regulation using gated self-attention layers. The normalized interaction frequencies (see Supplementary Method 1) are injected to self-attention score matrices to guide the model with the priorities of interactions. Detailed explanation of the model is illustrated in Methods and Supplementary Note 1.

## Chromoformer outperforms existing deep models for gene expression prediction based on epigenetic features

We benchmarked the performance of Chromoformer with three baseline deep-learning models using the optimal settings proposed in the respective studies. We first trained DeepChrome[2], a convolutional neural network that learns the local combination of HMs through the weights of convolutional filters to predict gene expressions. We also trained AttentiveChrome[3] and DeepDiff[21] models. The former combines LSTM and global attention mechanism to enhance the interpretability of the model, and the latter extends it to predict the fold-change of gene expression between a pair of cell types. Lastly, an HM-based hybrid CNN-RNN model proposed by Kang et al.[22] (HM-CRNN) was also chosen for the comparison. It learns and captures the meaningful local combination of HMs through CNN and comprehends their sequential dependencies through RNN.

The predictive performance of Chromoformer was comprehensively evaluated on three different gene expression prediction tasks: Binary gene expression state classification, expression level regression, and expression fold-change regression. In the binary gene expression state classification task, model predicts whether an expression level of a gene is above median or not. This problem formulation was first proposed by Singh et al.[2], and it has been so far widely adopted for many studies, including the three aforementioned baseline studies. In the expression level regression task, models were trained to predict log2-transformed RPKM values, and the expression fold-change regression task evaluates the performance to predict the expression fold-change between the two cell types for each gene. For each task, we designed a tailored variant of Chromoformer models (Fig. 2a), all of which are based on the multi-scale backbone of Chromoformer (Fig. 1b–e). Chromoformer-classifier (Chromoformer-clf) was built for the binary gene expression state classification task. It has a classification head with fully-connected layers that produces a two-dimensional probability vector denoting the probability of high or low expression.

Chromoformer-regressor (Chromoformer-reg) was designed for the expression level regression task using a regression head producing a single scalar. Chromoformer for fold-change regression (Chromoformer-diff) adopts a Siamese neural network architecture to accept HM profiles from two different cell types (Supplementary Fig. 5a). The two Chromoformer backbone share their weights, so that the two HM profiles can be embedded in the same latent space and their differences can be nonlinearly translated into fold-change value by the subsequent regression head.

Model performances were evaluated for 11 cell types among the 127 cell types profiled by Roadmap Epigenomics[23] and ENCODE project[24]. Those 11 cell types were chosen because all of the gene expression profiles, ChIP-seq data for seven major HMs, and pcHi-C interaction profiles were publicly available for each of the cell types. A total of 18,955 genes were split into four sets for 4-fold cross-validation (CV), each consisting of 5045, 4751, 4605, and 4554 genes. For every CV fold, each set became a held-out validation set, while the other three sets were used for model training. To avoid unwanted information leakage from the training to validation set through 3D chromatin folding involving promoter-promoter interaction, we ensured that no two genes in different sets are located on the same chromosome.

As a result, our multi-scale Chromoformer model achieved significant performance improvement over existing baseline deep learning models in all three tasks, suggesting that the proposed model architecture was successful in modeling the regulatory hierarchy of gene expression (Fig. 2b–f and Supplementary Figs. 5b, 6a, b and 7). These results were consistently reproduced for all 11 cell types examined. In detail, for the binary gene expression state classification task, Chromoformer-clf achieved significant performance improvement over existing baseline deep learning models in terms of area under receiver operating characteristic curve (ROC-AUC) (Fig. 2b), accuracy, and average precision (Supplementary Fig. 6a, b). Besides, we found that the prediction probabilities produced by Chromoformer-clf showed a very high positive correlation with the actual expression levels (Supplementary Fig. 6c). These well-calibrated prediction probabilities for the quantitative expression levels support the use of binary classification formulation for the quantitative modeling of HMs. Chromoformer-clf also far outperformed GC-MERGE, a GNN using three-dimensional chromatin interaction to predict gene expression (Fig. 2c). Importantly, GC-MERGE can only predict for genes involved in at least one chromatin interaction. Also, GC-MERGE can only predict the gene expression in the unit of 10 kbp genomic bins, therefore it cannot produce gene-wise prediction when two or more genes are present in the same bin. Therefore, Chromoformer was retrained from scratch for a subset of genes whose expression can be predicted by GC-MERGE for a fair comparison. Meanwhile, Chromoformer-reg outperformed the regression variants of benchmark models in terms of Pearson's correlation coefficient (Fig. 2d) and $R^2$ (Supplementary Fig. 7). Chromoformer-diff also had significantly better performance over the state-of-the-art model DeepDiff (Fig. 2e, f). Of note, it was also much better than the regression performance using the ratio of prediction probabilities of classification models as predicted fold-changes (Fig. 2e). Collectively, these results show the effectiveness of Chromoformer architecture in epigenetic gene regulation prediction.

## Training with large window size and *cis*-regulatory interactions contributed to the performance improvement in Chromoformer

To dissect the performance of Chromoformer into the contributions of individual factors, we first inspected for the effect of modeling wide-range windows around the TSS up to 40 kbp. By gradually increasing the window size around TSS from 2 kbp to 40 kbp, we observed a consistent performance increase for our model, while other deep learning models showed considerable performance degradation when the window size was larger than 10 kbp (Fig. 3a). Larger windows are

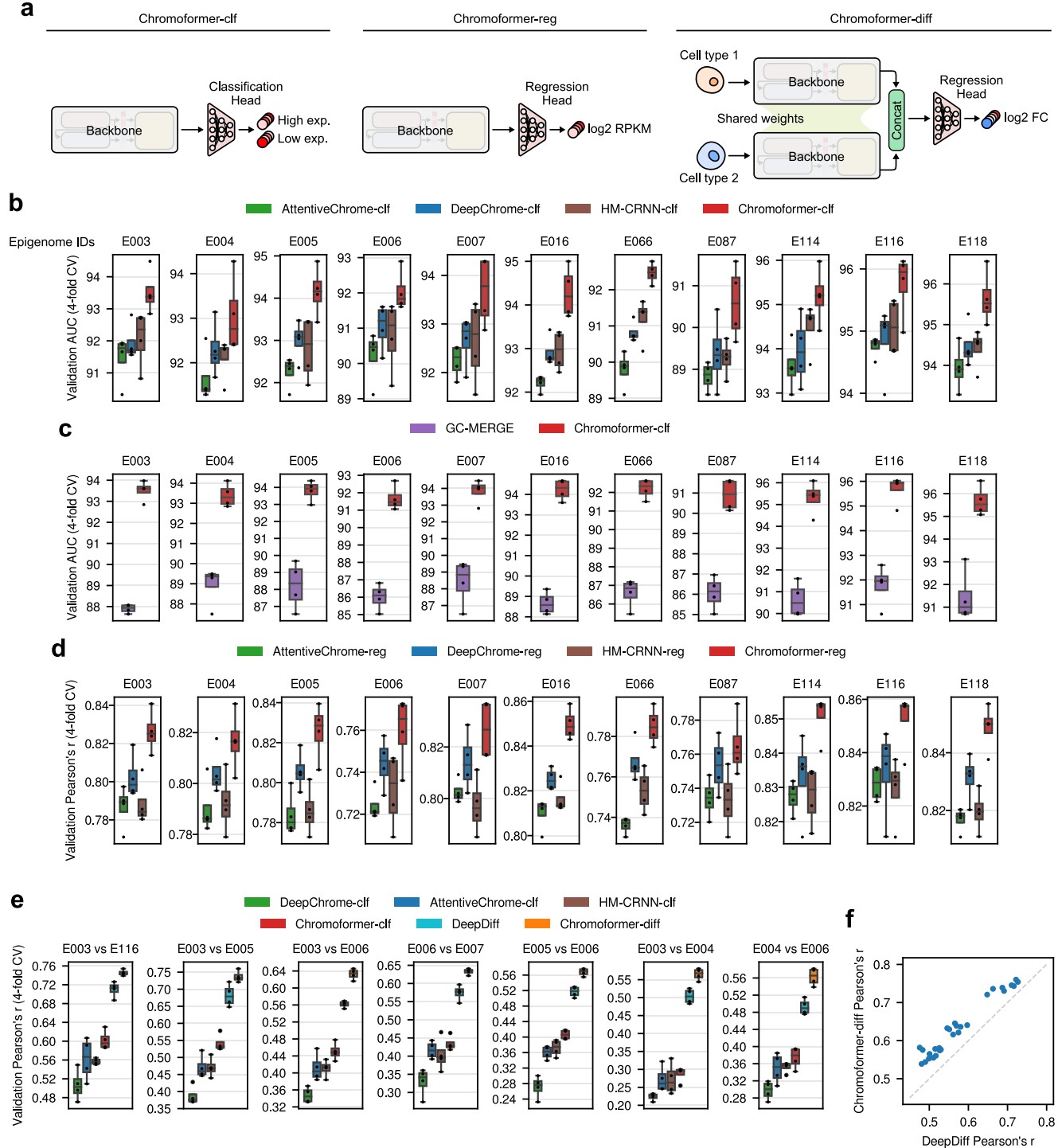

**Fig. 2 | Chromoformer outperformed existing deep learning models in predicting gene expression in various aspects. a** Three variants of Chromoformer evaluated in this study. Chromoformer-classifier (Chromoformer-clf) predicts binary gene expression labels. Chromoformer-regressor (Chromoformer-reg) predicts the expression levels (in log2 RPKM) of genes. Chromoformer-diff predicts the log2 fold-change of gene expression given the HM configurations of two different cell types for each gene. **b** Cross-validation (*n* = 4) performances of Chromoformer-clf compared to the benchmark deep learning models that only utilize the core promoter features. **c** Comparison of cross-validation (*n* = 4) performances with GC-MERGE, a graph neural network model that utilizes 3D *cis*-regulatory interactions. For fair comparisons, Chromoformer-clf models were retrained from scratch using only a subset of genes that GC-MERGE can predict.

**d** Cross-validation (*n* = 4) performances of Chromoformer-reg compared to the benchmark deep learning models. Prediction heads of benchmark models were modified to produce single scalar values instead of binary labels. **e** Cross-validation (n=4) performances of Chromoformer-diff compared to the benchmark deep learning models. Evaluation of the fold-change prediction based on the ratio of classification probabilities are shown as a reference (denoted as DeepChrome-clf, AttentiveChrome-clf, HM-CRNN-clf, and Chromoformer-clf). **f** Pairwise performance comparison between DeepDiff and Chromoformer-diff. Throughout **b**–**e** the center line denotes the median, the upper and lower box limits denote upper and lower quartiles, and the whiskers denote 1.5× interquartile range. AUC, Area under the receiver operating characteristic curve; CV, Cross-validation. Source data are provided as a Source Data file.

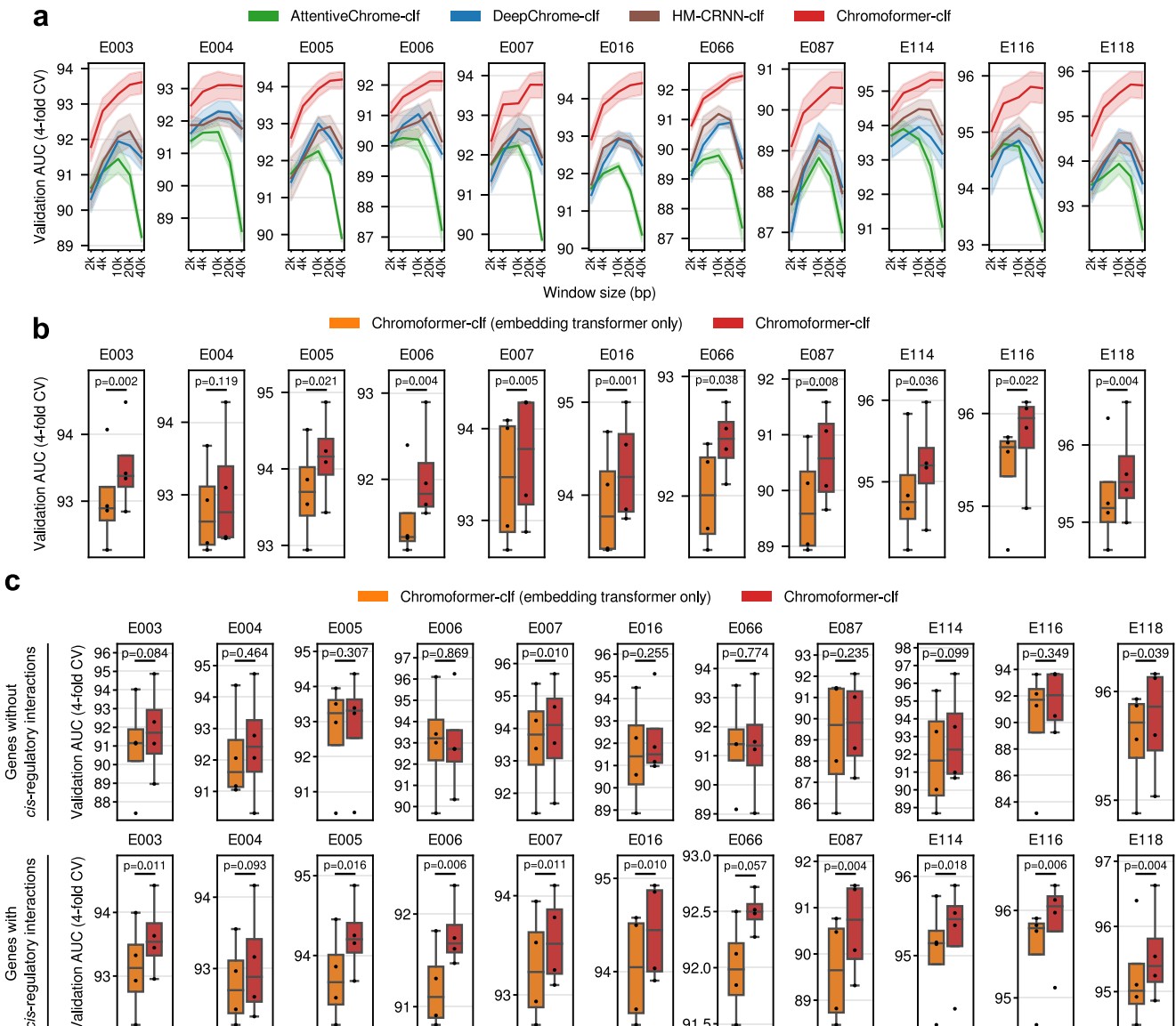

**Fig. 3 | Contributing factors to the superior performance of Chromoformer.**
**a** Effect of input window size around TSS on the model performance. Chromoformer and the other benchmark models were trained for five different window sizes (2 kbp, 4 kbp, 10 kbp, 20 kbp, and 40 kbp), while all the other training procedures were kept the same as previously. Bold lines denote the average validation AUC across 4-fold cross-validation for each window size, while the shades denote the standard error of them. **b** Effect of taking distal *cis*-regulations by pCREs into account. We trained ablated Chromoformer models which only have the Embedding transformer and thus cannot incorporate the *cis*-regulatory information between the core promoter and pCREs. The resulting cross-validation ($n = 4$)

performances were compared with the intact Chromoformer model. **c** Comparison of the cross-validation ($n = 4$) performances for a subset of genes without or with known chromatin interactions. ROC-AUC scores of Embedding transformer-only Chromoformer and intact Chromoformer were computed only for a subset of genes that do not have known *cis*-regulatory interactions (Upper), and genes with at least one known *cis*-regulatory interactions. *P*-values from two-sided paired *t*-tests are shown. In the boxplot, the center line denotes the median, the upper and lower box limits denote upper and lower quartiles, and the whiskers denote 1.5× interquartile range. Source data are provided as a Source Data file.

more likely to include TSSs of other genes, which jeopardizes the model performance since the model will be more prone to the spurious attention towards those irrelevant TSSs. This is because the attention score for each genomic position is computed against global context vectors without accounting for the absolute distance to the target TSS. In the case of the transformer architecture, however, scaled-dot product attention with positional encoding allows the computation of attention scores among all pairs of positions without introducing a global context vector, thereby allowing us to query which position within the input window the specific TSS of our interest is attending on. Thus, these results emphasize the strength of the transformer architecture in pinpointing histone codes that are relevant to the regulation of gene expression within a wide-range window around the TSS.

We next examined whether the incorporation of distal *cis*-regulatory interactions into modeling truly contributed to the performance improvement. To this end, we eliminated Pairwise Interaction and Regulation transformers from the Chromoformer model to see how well it performs when trained without using distal *cis*-regulatory interactions. We retrained the ablated Chromoformer models and compared their performances to the intact Chromoformer model in terms of ROC-AUC (Fig. 3b). This revealed that the Chromoformer in its intact form showed significantly high performances in most of the analyzed cell types (10 out of 11), implying that the inclusion of distal pCREs and their interactions into deep modeling helped learning epigenetic factors that govern the expression of genes. To further support that this improvement is specifically due to the modeling of

*cis*-regulatory interactions, but not due to the mere complexity of the model, we investigated the modeling performance separately for the genes whose promoters do not interact with any pCREs. Since no biologically meaningful information would be transferred to the core promoter embeddings of those genes through the Pairwise Interaction and Regulation transformers, we can discern whether the dominant factor that contributed to the performance increase was the information transfer between genomic regions or the increased model complexity itself. As expected, we observed that Chromoformer showed no significant improvements for the majority of cell types (Fig. 3c). Besides, the performance increase was still significant for the genes having at least one interaction with pCREs (Fig. 3c). Furthermore, the contribution of intra-TAD interactions to the performance were shown to be greater than inter-TAD interactions, again supporting the biological relevance of distal *cis*-regulatory interactions in Chromoformer training (Supplementary Fig. 8). Together, it indicates that the contribution of *cis*-regulatory modeling is far greater than that of increased modeling capacity arising from deeper layers.

## Chromoformer learns to attend to the distant transcriptional elongation signals at gene bodies

Given the improved performance of the Chromoformer model compared to the other deep learning models, as well as the success of the wide-range modeling of core promoter region, we then asked if there are any unique patterns of HMs captured by Embedding transformers. As the self-attention layers inside the embedding transformers were designed to comprehend the dependencies of HMs at core promoters, we postulated that any such dependencies that the Chromoformer model is aware of can be revealed through the self-attention map it yields. Therefore, the attention weights produced by the Embedding transformer of Chromoformer-clf during the prediction were visualized to analyze the internal behavior of the model. Figure 4a shows an example snapshot of attention weights during the prediction of the expression of anti-silencing function 1 A (*ASF1A*) histone chaperone in the H1 cells (Epigenome identifier E003). Interestingly, we observed that a majority of attention heads were prominently giving strong attention to 4-6 kbp downstream the TSS than any other specific positions (Fig. 4a, b). This was rather unexpected since most of the regulatory information delivered by HMs are known to be deposited near the TSS, where the binding of transcription factors and transcriptional initiation mostly take place. In line with this notion, average signals of HMs displayed their characteristic patterns mostly at the TSS (Fig. 4c). Specifically, methylations at H3K4 and histone acetylations were associated with higher levels of gene expressions, whereas high levels of H3K9me3 and H3K27me3 modifications were associated with low expression of genes.

In striking contrast, the average H3K36me3 signal of the two classes of genes (i.e., high/low expression) did not show any notable difference at the TSS, but the difference was maximized at the 4–6 kbp region downstream the TSS (Fig. 4d). H3K36me3 is established by the addition of methyl groups at H3K36 by SETD2 histone methyltransferase, and SETD2 is known to be recruited to the C-terminal domains of RNA polymerases in concert with the transcriptional elongation[25]. Thus, H3K36me3 is widely appreciated as a transcriptional elongation-associated HM that predominantly marks the bodies of actively-transcribed genes. The average attention weights imposed by the TSS-containing bin reached its maximum exactly at 4-6 kbp region downstream the TSS (Fig. 4e), suggesting that the model was well optimized to focus on the most discriminative genomic region in terms of H3K36me3 signals. Intriguingly, the extent of the attention given at the 4–6 kbp region downstream TSS was far greater for genes with high expression than low expression (Fig. 4e). In other words, the Embedding transformer was trained to adaptively control the amount of attention given on the HMs at the gene body, based on the histone context at the direct vicinity of the TSS, as illustrated in Fig. 4f. These

patterns of self-attention weights were highly consistent across the cell types examined (Supplementary Fig. 9). One explanation for this behavior of the Embedding transformer is that the model seeks for the complementary evidence that reinforces its confidence for the initial guess on the gene expression, which is based on the HM states near the TSS. In this sense, H3K36me3 is a well-suited candidate for such a role since its discriminative power resides where the other HMs do not show large variabilities, hence being left as the only clear signal in those regions. The importance of H3K36me3 at downstream gene bodies was further supported by the feature ablation experiment. When the H3K36me3 signals were excluded from model training and only the other six HMs were used as input features, we observed a significant decrease in performance (Fig. 4g). Moreover, we discovered that the Embedding transformer predominantly lost specific attention to the 4–6 kbp downstream region (Fig. 4h).

Notably, H3K36me3 ablation was the most detrimental in model performance compared to the ablation of any other individual HMs (Supplementary Fig. 10a). This implies that the distribution of H3K36me3 may not be readily inferred by the other HMs, as shown by its low spatial correlation with them (Supplementary Fig. 10b). Furthermore, while ablating the combinations of other HMs corresponding to the chromHMM-defined 50 chromatin states generally resulted in poor performance, the effect of H3K36me3 almost seemed to be independent (Supplementary Fig. 10c, d). We also noticed that ablation of the two enhancer marks, H3K4me1 and H3K27ac, was not solely sufficient to significantly degrade the performance of Chromoformer, while further ablating H3K4me3 and H3K9ac incurred drastic performance drop (Supplementary Fig. 10c). This implies the spatial correlation or the redundancy of active HMs (Supplementary Fig. 10b) were effectively compensating the absence of the regulatory information conveyed by other active HMs. In addition, it may be due the compensation through promoter-promoter interactions between active promoters marked by H3K4me3 or H3K9ac that hint the existence of the transcriptional hubs enriched with enhancers. Nevertheless, the ablation of H3K4me1 and H3K27ac with additional HMs generally showed high-performance degradation without H3K36me3 ablation, suggesting the overall importance of HMs marking enhancers.

It seems that the importance of the feature representing transcriptional elongation was particularly important in this problem setting because the model was trained to predict the steady-state levels of mRNA measured by RNA-seq. Steady-state levels of mRNA are determined not only by the transcriptional initiation, but also by various factors, including the rate of transcriptional elongation and mRNA stability. According to a study comparing different measurement technologies used for gene expression prediction tasks[26], predicting the expression levels measured by cap analysis of gene expression (CAGE) was shown to be easier than predicting RNA-seq-based expression levels. This study also showed that H3K36me3 was predictive for RNA-seq-based expression levels, while core promoter HMs including H3K4me3 were more useful for CAGE measurements. They imply that the hidden factors, including the efficacy of transcriptional elongation, reside in RNA-seq measurements, and they cannot be readily accounted for with core promoter features alone. Thus, we speculate that the superior performance of Chromoformer model may arise from the ability to model the rate of transcriptional elongation, which leaves its trace as H3K36me3 in the gene body. These results, based on the great interpretability of the Embedding transformer, collectively suggest that the Embedding transformer learned the distant correlation between histone codes dictating active transcription near TSS and high levels of H3K36me3 representing transcriptional elongation at gene body, especially at the gene bodies 4-6 kbp downstream the TSS. Moreover, this in part explains why the performance of Chromoformer showed consistent increase along the increase of window size, as the model could collect additional evidence for gene expression also from the gene body.

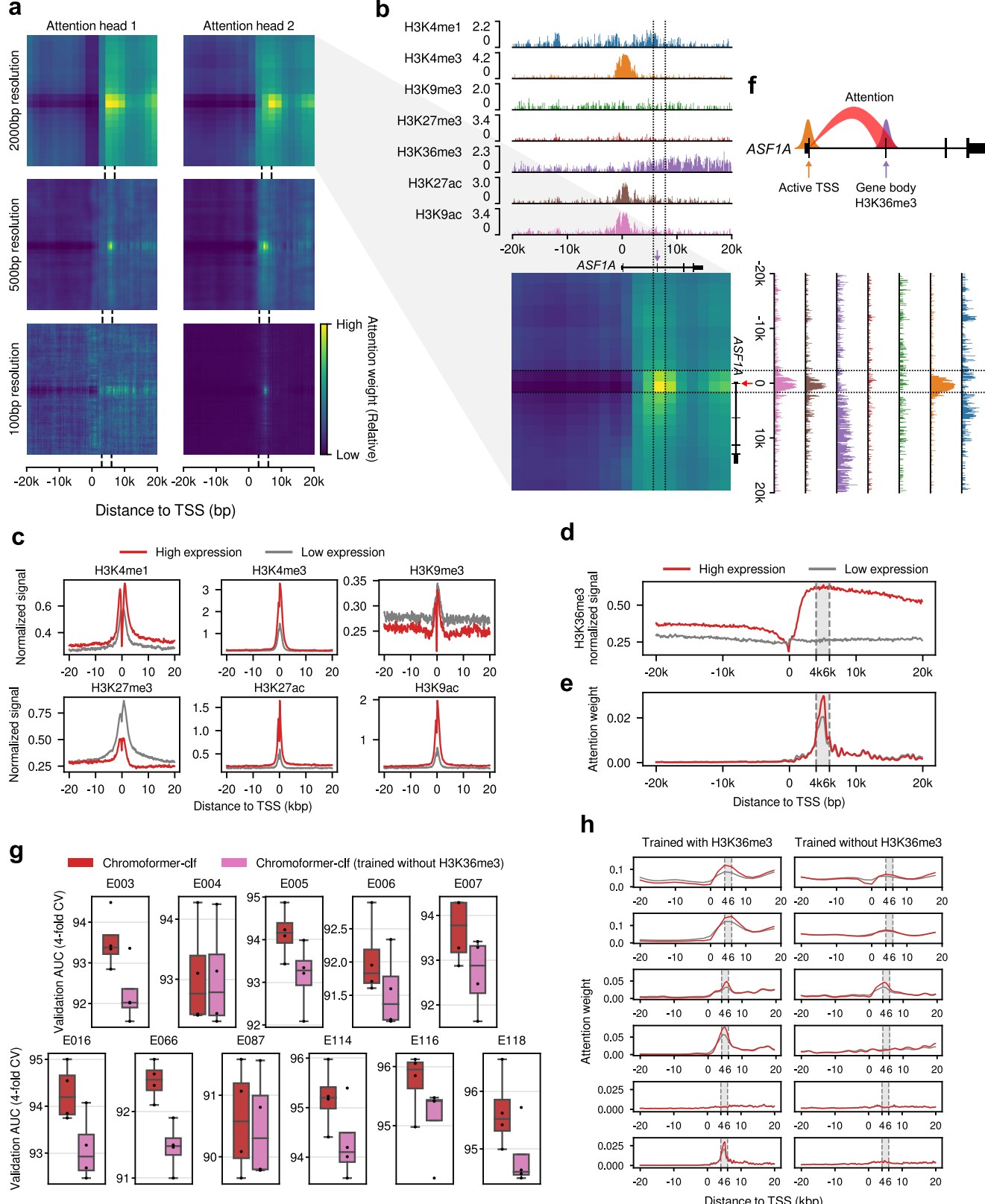

## Analyzing *cis*-regulome through *cis*-regulatory impact predicted by Chromoformer

We then examined the effect of modeling *cis*-regulation by distal pCREs in detail. As it has already been shown that the inclusion of Pairwise Interaction and Regulation transformers results in better overall performance, we sought for a more detailed interpretation

for the gene-level effect of regulatory interaction modeling. To this end, we devised a gene-level measure to quantify the predicted impact of the *cis*-regulation based on the latent representation learned by Chromoformer. Specifically, we measured the Euclidean distance between the interaction-aware regulatory embedding produced by Pairwise Interaction and Regulation transformers, and

**Fig. 4 | Analysis of self-attention weights learned by the Embedding transformer. a** Representative self-attention weight matrices for the prediction of the expression of ASF1A in H1 cell. Each heatmap shows the attention weight for each pair of genomic bins. The dotted lines indicate the 4–6 kbp region downstream TSS. **b** Detailed description of the attention weights learned by the attention head 2 for 2 kbp-resolution Embedding transformer. Genome tracks representing the normalized signals of the seven HMs are aligned with the attention weight matrix. The dotted lines demarcate the regions 4–6 kbp downstream the TSS. The red arrow indicates the TSS, and the purple arrow indicates the exon located within the region 4–6 kbp downstream the TSS. **c** Average signals of the HMs other than

H3K36me3. Signals were separately averaged according to their expression labels (High/Low expression). **d** Average H3K36me3 signal. **e** Average attention weights of the second attention head for 100 bp resolution bins. The grey shade denotes the 4–6 kbp region downstream the TSS. **f** Schematic diagram illustrating the behavior of the Embedding transformer. **g** Cross-validation ($n = 4$) performances upon H3K36me3 feature ablation. The center line denotes the median, the upper and lower box limits denote upper and lower quartiles, and the whiskers denote 1.5× interquartile range. **h** Attention weights for TSS bin. Red and grey lines indicate the average attention weights for genes having above-median and below-median expression, respectively. Source data are provided as a Source Data file.

the original core promoter embedding resulting from Embedding transformers (Fig. 5a). For convenience, we termed this quantity as 'predicted *cis*-regulatory impact (PCRI)'.

To assure that PCRI truly reflects the dynamics of latent vector representations as well as the effect of regulatory interactions, we first asked how it eventually affected the prediction outcome probabilities. The fold-change in predicted gene expression between the interaction-aware Chromoformer-reg and interaction-free Chromoformer-reg was measured for each gene to determine the amount of perturbation on the prediction probability. As a result, fold-changes for the two groups of genes with above and below half PCRI were significantly different within each of the low expression and high expression groups of genes in H1 cells (E003) (Fig. 5b). Similarly, PCRI and fold-change for the highly expressed genes were positively correlated (Spearman's $r = 0.39$, $p < 10^{-308}$) and PCRI and fold-change for the lowly expressed genes were negatively correlated (Spearman's $r = -0.54$, $p < 10^{-308}$) (Fig. 5c). In short, it can be summarized that high PCRI for highly-expressed genes made those genes predicted with higher gene expression, and high PCRI for lowly-expressed genes resulted in lower predicted expression.

To get deeper insights into how the Chromoformer model could accurately discern pCREs associated with the activation or suppression of gene expressions, we analyzed the characteristics of pCREs assigned for genes at the highest extreme in terms of PCRI, i.e., genes predicted to have greatest impact by distal *cis*-regulation. We collected 250 highly-expressed genes with the highest PCRI values from each fold of the 4-fold CV and examined the average signals of HMs near the pCREs associated with those 1000 genes. As a result, pCREs for highly-expressed genes with high PCRI on average showed increased levels of HMs associated with transcriptional activation compared to those for lowly-expressed genes (Fig. 5d, shown for H1 cells). In particular, HMs representing enhancers (H3K27ac, H3K9ac, and H3K4me1), active promoters (H3K4me3 and H3K9ac) and active gene bodies (H3K36me3) were enriched for those pCREs. This broad enrichment of genomic elements associated with the greatest transcriptional activation implies that Chromoformer learned the existence of transcription factories, on which the active genes and enhancers are gathered together for efficient transcription[27]. Based on this observation, we sought for additional biological evidence by examining whether those genes clustered at putative transcription factories show enrichment for particular biological functions. Interestingly, they were highly enriched for housekeeping activities, including mRNA splicing, DNA replication, ribosome biogenesis, and DNA damage response (Fig. 5e). We also observed the enrichment of cell type-specific functions such as telomere maintenance in stem cells (E003 and E016) and immortalized cell lines (E114, E116, and E118), cell morphogenesis and extracellular structure organization in mesenchymal stem cells (E006) or iron homeostasis in liver cells (E066) and hepatocellular carcinoma (HCC) (E118) (Supplementary Fig. 11). Taken together, it can be speculated that Chromoformer reflected the tendency of cells to ensure the robust expression of essential genes for its function and survival through sequestering them

within transcriptionally active subcompartments harboring multiple enhancers[28]. On the other hand, pCREs for lowly-expressed genes with high PCRI on average showed increased levels of repressive marks such as H3K27me3 and H3K9me3 (Fig. 5f), implying that Chromoformer also detected the transfer of the suppressive regulatory information from the pCREs to the core promoter. We conjectured that those pCREs represent transcriptional silencers, since previous studies have shown the potential functionality of distal H3K27me3-rich regions in transcriptional repression[29]. The top 1000 genes which are predicted to have strong suppressive *cis*-regulation showed extreme enrichment towards developmental functions (Fig. 5g). One representative example of the suppressive *cis*-regulatory interactions is shown in (Fig. 5h) for Engrailed Homeobox 2 (*EN2*). As expected, many of the pCREs showed high H3K27me3 signals. Notably, one of the pCREs was located 1.5 Mbp away from *EN2*, and the pCRE spanned the core promoter of Motor Neuron and Pancreas Homeobox 1 (*MNX1*), which is another homeobox transcription factor associated with development. The functional similarity between the two highly distant, but interacting genes implies the existence of silencing hubs, where the developmental genes and silencers are sequestered together through 3D chromatin folding.

Assuming that the collection of the whole PCRI values can be regarded as a *cis*-regulome of each cell type, we further asked whether we can detect differential *cis*-regulation between cell types and unveil its underlying basis in terms of histone codes. We selected top 1000 genes having highest variances of normalized PCRI (Methods) across the 11 cell types used in this study (Supplementary Fig. 12) and performed hierarchical clustering with their PCRI values. As expected, similar cell types were clustered together and corresponding cell-type specific functions were highlighted by high PCRI values (Fig. 6a). Notably, we observed that the *cis*-regulomes of healthy liver tissue and HepG2 HCC cells were tightly clustered, but also found a small subset of genes that were being subjected to HepG2-specific *cis*-regulation (Fig. 6a, black box). We could not find any biological terms significantly enriched for these genes unlike other clustered gene sets in the analysis, so it could be speculated that they represent a consequence of cancer-specific aberrant cis-regulation occurring in a stochastic manner. In support of this notion, we could identify four individual genes (*GNA12*, *TRIB3*, *CCN2* and *RBM39*) tightly implicated in HCC[30–34], which can be thought as epigenetic hits by aberrant cis-regulation. The expression of the four genes were 9.3-, 6.0-, 4.1-, and 3.7-fold higher in HCC than in healthy liver cells, respectively, in accordance with the tendency of PCRIs (Supplementary Fig. 13). To interpret why the Chromoformer predicted high PCRI values for those genes, we visually inspected the histone modification landscape surrounding the genes. For example, Fig. 6b shows histone modification landscapes around *CCN2* in healthy liver and HCC cells. Comparing the two landscapes revealed a putative enhancer region that is only active in HCC (Fig. 6b, red arrow), which may explain higher PCRI as well as higher expression of *CCN2* in HCC. It highlights that the in-depth interpretation of Chromoformer model prediction in the

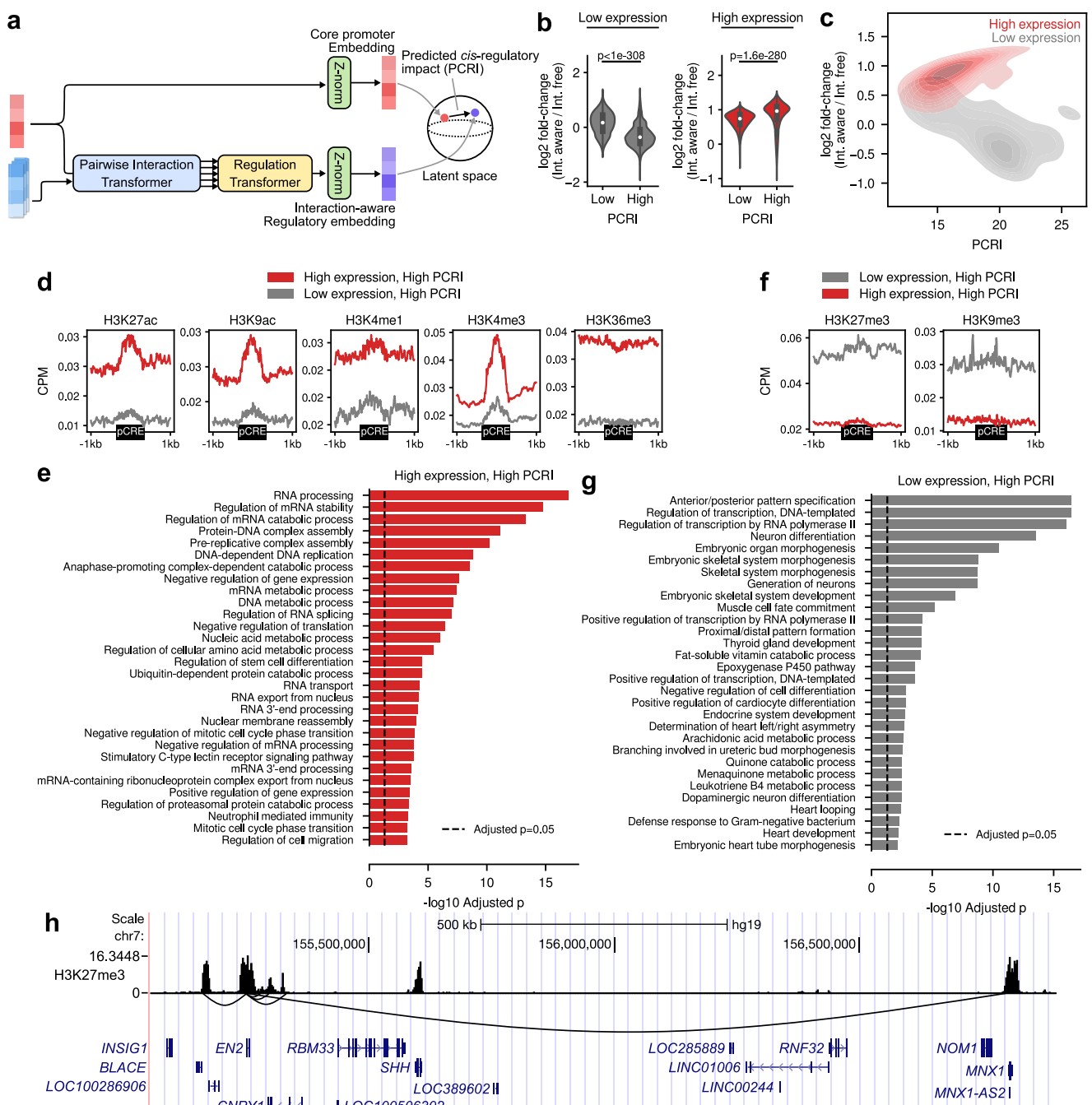

**Fig. 5 | Analysis of predicted *cis*-regulatory impact (PCRI). a** Schematic diagram illustrating the computation of the PCRI. **b**, **c** Relationship between PCRI and the log2 fold-change of predicted gene expression between the interaction-aware and interaction-free Chromoformer-reg in H1 cells (Epigenome identifier E003). Low and high PCRI groups were split by the median PCRI value. In **b**, *p*-values from two-sided Wilcoxon rank-sum tests between genes with below-median PCRI ($n = 4739$ for both low and high expression groups) and above-median PCRI ($n = 4738$ and $4739$ for low and high expression groups, respectively) are shown. In the boxes within the violinplot, the white point denotes the median and the upper and lower box limits denote upper and lower quartiles. **d** Average HM signals near the pCREs interacting with top 1000 genes with the highest PCRI among highly and lowly expressed genes. **e** Functional enrichment analysis of the top 1000 genes with the

highest PCRI among highly expressed genes in H1 cells. Benjamini-Hochberg adjusted Fisher's exact *p*-values are shown in negative logarithmic scale. **f** Average HM signals near the pCREs interacting with top 1000 genes with the highest PCRI among highly and lowly expressed genes. **g** Functional enrichment analysis of the top 1000 genes with the highest PCRI among lowly expressed genes in H1 cells. Benjamini-Hochberg adjusted Fisher's exact *p*-values are shown in a negative logarithmic scale. **h** Representative genomic region showing suppressive *cis*-regulatory interactions for *EN2*. Black curved lines below the H3K27me3 signal track shows the 3D chromatin interactions centered at the core promoter of *EN2*. NCBI RefSeq gene annotations are shown at the bottom. Source data are provided as a Source Data file.

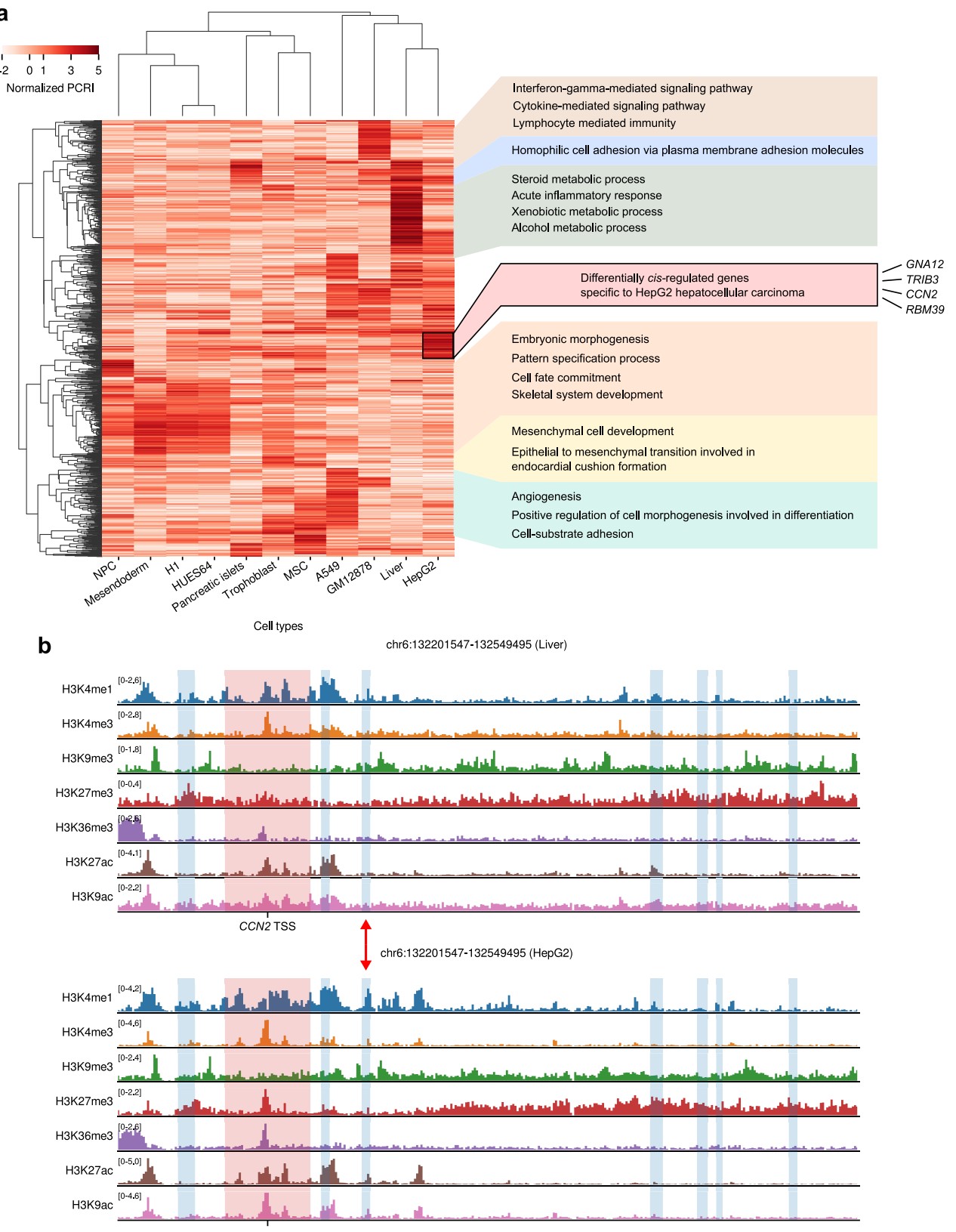

**Fig. 6 | Differential *cis*-regulome analysis using PCRI values. a** Hierarchical clustering of top 1000 genes having the highest normalized PCRI variances across cell types. Representative GO terms enriched for the corresponding set of genes are shown on the right. **b** Histone modification landscape around the transcription start site of CCN2 and its pCREs in Liver (E066) and HepG2 (E118) cells. Red shades denote promoter regions and blue shades denote pCRE regions interacting with the promoter. Red arrow represents a putative enhancer region that seems to be only active in HepG2 cells. Source data are provided as a Source Data file.

form of differential cis-regulome analysis can reveal an epigenomic origin of malignant gene expression.

## Chromoformer learns the additive transcriptional activation at transcription factories and switch-like suppression at Polycomb-bound silencing hubs

To assess the quantitative characteristics of the two regulatory hubs, namely transcription factories, and silencing hubs, we analyzed the correlation between the levels of PCRIs and the number of pCREs associated with transcriptional activation or suppression. First, to define transcriptionally active regions across the genome, we utilized the chromHMM-based chromatin state annotations available on the Roadmap Epigenomics Project. Among the 18 states, genomic intervals representing active TSS, active gene bodies or enhancers were considered as active regions, and we counted for each gene the number of pCREs overlapping with the identified active regions. We found that the number of the transcriptionally active pCREs and PCRI showed a moderate but significant positive correlation (Spearman's $r = 0.15$, $p = 1.3 \times 10^{-42}$) in H1 cells (Fig. 7a). As the expression levels of genes also increased in accordance with the increasing number of transcriptionally active pCREs (Fig. 7b), it implies that Chromoformer learned the additive dynamics of pCREs in transcription factories for gene expression activation (Fig. 7e).

For silencing hubs, we considered the pCREs associated with Polycomb group (PcG) proteins as functional silencers. Besides its enzymatic role as a histone methyltransferase targeting H3K27, it has been recently demonstrated that Polycomb repressive complex 2 (PRC2) functions as a mediator of the repressive cis-regulatory interaction between silencers and developmental promoters, promoting the formation of PcG bodies[35]. To determine the number of PRC2-bound silencers, we utilized the ChIP-seq peaks for the two of the three core subunits of PRC2, namely EZH2 and SUZ12, from ENCODE. As a result, we found the logistic increase of PCRI upon the increasing number of pCREs overlapping with the EZH2 binding site in H1 cells (Fig. 7c, d), which indicates that the switch-like transcriptional suppression (Fig. 7f) by PRC2-mediated silencing hub was learned by Chromoformer. Similar results could be reproduced for SUZ12 binding (Supplementary Fig. 14a, b). Notably, this trend entirely disappeared when tested for the number of non-specific pCREs irrelevant to PRC2 (Supplementary Fig. 14c), which is suggestive of the highly specific nature of PRC2-mediated silencing. We further asked in which cell types, other than H1, the switch-like dynamics of PRC2-mediated silencing are observed. As a result, we found that such patterns in mesendoderm, neural progenitor cells and HUES64 cells (Supplementary Fig. 15). Interestingly, these cell types were already shown to have close similarities when the various cell types were clustered based on their H3K27me3 mark in repressed Polycomb chromatin states[23]. Thus, our results highlight that the similarities in 1D epigenomic states, especially Polycomb-associated H3K27me3 mark, can be recapitulated by the similarities in the functional impact of cis-interactions. We also examined if the pCREs associated with Polycomb repressive complex 1 (PRC1) has different patterns of gene silencing and observed similar patterns (Supplementary Fig. 16). This may reflect the cooperativity of PRC1 and PRC2 in their localization at Polycomb response elements[36], but we cannot exclude the possibility that the model missed some genes that were subjected to PRC1- or PRC2-specific cis-silencing.

To date, the collective effect of distal cis-regulatory elements on gene expression remains incompletely understood, but nevertheless, the pioneering works exploiting modern technologies such as STARR-seq[37] or CRISPRi-FlowFISH[38] certainly provide us with deep insights about their dynamics. Intriguingly, the observations drawn from the interpretation of trained Chromoformer models, which are optimized to capture the quantitative characteristics of cis-regulation, highly agree with the latest viewpoints from such studies. Our observations on the additive transcriptional activation by active pCREs recapitulates the results of a previous study on the quantitative characterization of enhancer activity in *Drosophila*. The underlying mechanism for this additivity has been explained by either of the interaction hub or promoter competition model[39]. The former assumes multi-way interactions between a promoter and several enhancers with independent contributions, while the latter posits the one-to-one promoter-enhancer interactions and demonstrates that the probability of contact between a promoter and any enhancer increases as the number of candidate enhancers increases. On the contrary, the quantitative nature of transcriptional silencing by PcG bodies with regard to the number of PcG-bound silencers is yet to be fully characterized.

Our interpretation of Chromoformer leads to the hypothesis that there exists a certain threshold of the local concentration of silencers for PcG bodies to fully exert their suppressive function (Fig. 7c). It may be due to the synergy with other repressive epigenetic factors, including the DNA methylation induced by the HMs newly added by those PcGs and other chromatin remodeling factors. In any case, experimental validation of this hypothesis and further characterization of the biological factors that determine the tipping point of the PcG-mediated gene silencing will highly improve our understanding of precise regulation of gene expression. Altogether, these results demonstrate the utility of Chromoformer and, by extension, deep learning models in the derivation of the new quantitative hypothesis in the field of computational biology that would ultimately facilitate experimental validations and thus new scientific discoveries.

## Discussion

In the present study, we proposed a transformer-based deep learning architecture named Chromoformer to model the quantitative role of the histone codes in the regulation of gene expression. Chromoformer greatly improved the performance of gene expression prediction by modeling the three-level hierarchy of cis-regulation involving core promoters and pCREs. By the analyses of self-attention weights, latent embedding dynamics, and several feature ablation studies, we also provided in-depth biological interpretations regarding the behavior of the Chromoformer model. Thanks to the power of transformers for comprehending distant dependencies in a sequence, Chromoformer could successfully learn to focus on the specific region inside gene bodies where the HMs associated with gene expression were the most distinctive between highly expressed and lowly expressed genes. Interestingly, the amount of attention paid to the gene body was dependent on the epigenetic context of the TSS, implying that the Chromoformer model captured the distant dependencies of the HMs placed at TSS and gene body. On the other hand, by using transformers to model pairwise relationships within an unordered set of features, Chromoformer could learn how the information mediated by histone code is propagated from pCREs to core promoters through 3D chromatin folding to regulate gene expression. Analysis of the latent representations of histone codes learned by the model highlighted that the expression of housekeeping and cell-type specific genes were reinforced by the interaction with enhancers, whereas the expression of developmental genes were mainly repressed by the interaction with PRC2-bound silencers.

We explicitly used a pre-compiled knowledge of 3D chromatin interactions to guide Chromoformer learning. Those experimentally measured interaction frequencies were used to prioritize the pCREs that will participate in the model training by being explicitly injected into the self-attention score matrices. However, it also seems possible to infer the interaction frequencies between pCREs and the core promoters from genomic sequence information alone. This is because the specificity of cis-regulatory interactions is largely governed by the recognition of DNA sequence motifs by DNA-binding proteins including transcription factors or CCCTC-binding factors (CTCFs), which function as insulators that compartmentalize the 3D genome conformations. Therefore, those binding motifs embedded in the

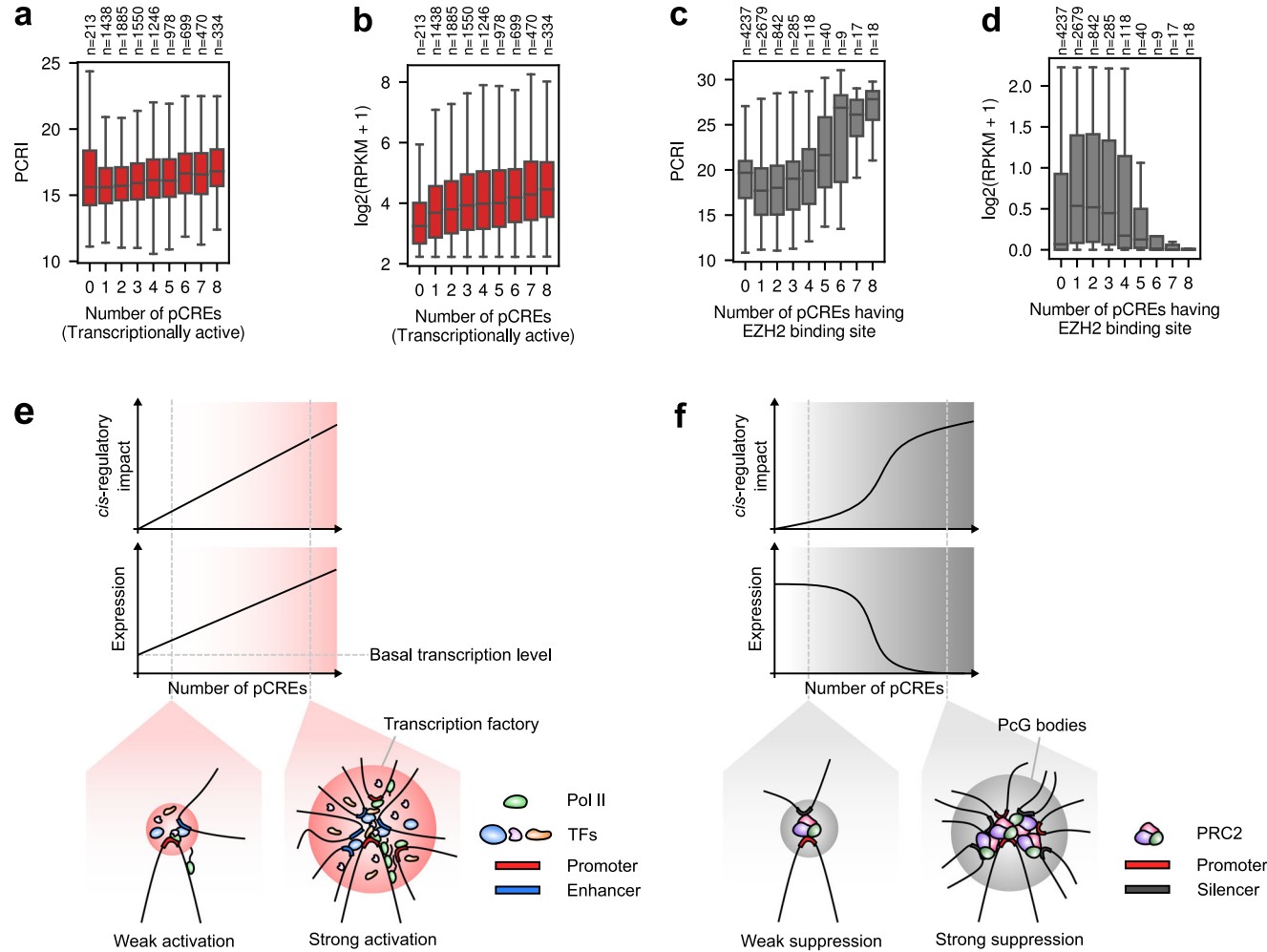

**Fig. 7 | Characteristics of *cis*-regulation learned by Chromoformer. a** Association between the number of transcriptionally active pCREs and PCRI. **b** Association between the number of transcriptionally active pCREs and gene expression level. **c** Association between the number of pCREs harboring EZH2 binding sites and PCRI. **d** Association between the number of pCREs harboring EZH2 binding sites and gene expression level. Throughout a-d, the number of genes having the corresponding number of pCREs are shown above the plot. In the boxplots, the center line denotes the median, the upper and lower box limits denote upper and lower quartiles, and the whiskers denote 1.5× interquartile range. **e, f** Illustrations for the proposed hypothetical models for regulatory dynamics of **e** transcription factories and **f** silencing hubs are shown. Pol II, RNA polymerase II; PcG bodies, Polycomb group bodies; PRC2, Polycomb repressive complex 2. Source data are provided as a Source Data file.

genome may serve as hidden vocabularies that allow the inference of the desired chromatin conformations solely based on the DNA sequence. Results from the recent model named Enformer strongly supports that such de novo prioritization of pCREs are more effective when wider sequence information is used[40], thereby suggesting the exciting possibilities for the fully data-driven modeling of gene expression regulation through the integration of genomic and epigenomic features using the transformer architecture. We leave this transformer-based multi-omics integration as a further work.

The attention learned by the Embedding transformer that jumps from an active TSS to the gene body suggests that the HMs placed at gene bodies are indeed useful, if not the most critical, information when predicting the steady-state gene expression levels. From this result, we can consider the possibility that using the entire landscape of histone codes distributed throughout a single gene may further improve the predictive accuracy for steady-state mRNA levels. Furthermore, as H3K36me3 is far more enriched at exons than introns, utilizing the full-length gene annotation will be another effective guidance for model training. As gene lengths and exon-intron distributions show great variability, we need some clever representation of such biological prior knowledge. Again, the transformer architecture would be one of the most powerful choices because one can flexibly

apply masks to deal with variable-length inputs and also can extend positional encoding to form composite encoding that simultaneously harbors information for both genomic positions and annotations for gene structures.

The proposed training scheme for Chromoformer models is highly expandable. For instance, we showed that Chromoformer models can be trained for cell types from species other than human, namely mouse embryonic stem cells, using relevant histone ChIP-seq[24] and Hi-C profiles[41], and the overall similarity between the grammars of histone codes between the two species was demonstrated through cross-species prediction performances (Fig. 8a–c). Also, cross-cell-type prediction experiments showed that a Chromoformer model trained in one cell type was still applicable to other cell types to some degree (relative validation AUC > 92%), with the cross-prediction performances being higher for similar cell types (Fig. 8d). This implies that Chromoformer trained in cell type-specific manner not only learned cell type-specific features of gene regulation, but also still captured the general rules that can be commonly applied to other cell types. Chromoformer training can be extended to incorporate any additional epigenomic feature if it can be represented as an array of genomewide signal values. Such features include transcription factor ChIP-seq signals or the first principal component (PC1) signals used for

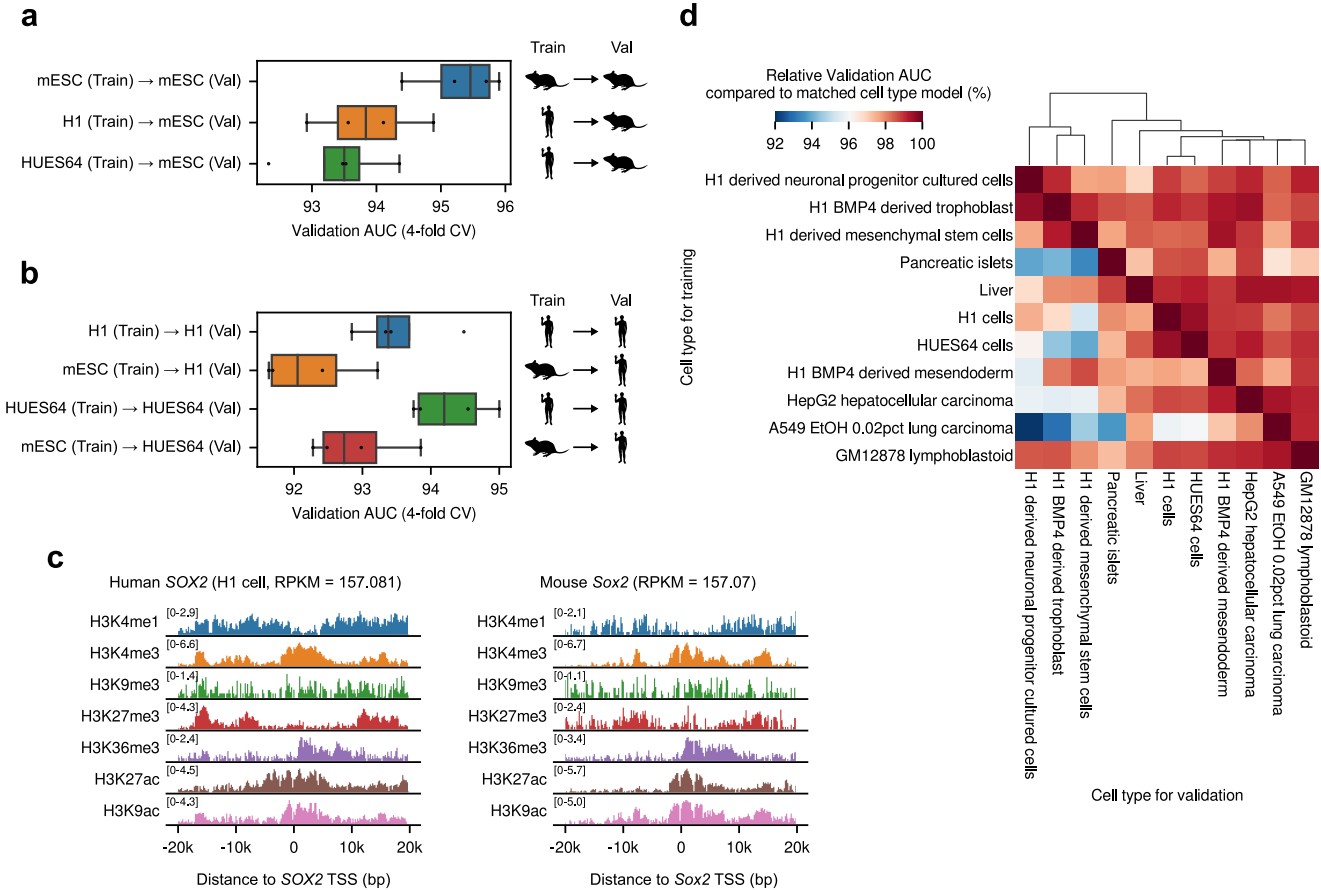

**Fig. 8 | Cross-species and cross-cell type prediction performances of Chromoformer. a, b** Cross-species prediction performances of Chromoformer-clf models. **a** Cross-validation ($n = 4$) performances of Chromoformer-clf models, trained with ES-Bruce4 mouse embryonic stem cells (mESC) or human embryonic stem cells (hESCs), are shown for the prediction of (**a**) mESC gene expression and (**b**) hESC gene expression. In the boxplot, the center line denotes the median, upper and lower box limits denote upper and lower quartiles, and whiskers denote 1.5× interquartile range. c Log2-transformed histone modification signals surrounding human *SOX2* TSS and mouse *Sox2* TSS. **d** Cross-cell type gene expression prediction performances. Colors represent the relative validation AUC compared to the matched cell type Chromoformer-clf model (i.e., trained and evaluated for the same cell type). Val, Validation. Source data are provided as a Source Data file.

compartment identification. As mentioned above, CTCF binding is a crucial determinant of 3D genome structure, and the promoter-proximal CTCF binding has been also highlighted in gene activation through distal enhancer-promoter interaction[42], as recapitulated in Supplementary Fig. 17a. We could show that including CTCF gave marginal but consistent increase in Chromoformer-clf performance (Supplementary Fig. 17b), and the increase was greater for Embedding transformer-only Chromoformer-clf model (Supplementary Fig. 17c). On the other hand, informing Chromoformer of cell-type specific genomic compartmentalization state[43] using PC1 value as an additional feature did not result in significant overall performance gain (Supplementary Fig. 18a). Even though the compartmentalization correlated with the levels of gene expression (Supplementary Fig. 18b, c), as the absolute level of the association (Pearson's correlation coefficient 0.12–0.19) is not great enough, we concluded that the predictive power of compartment-level features did not exceed that of gene-level HM features.

In summary, Chromoformer is another exemplary application emphasizing the great potential of transformer architecture in modeling biological sequences. This study also underscores the importance of developing specialized deep learning architectures effectively embedded with biological prior knowledge, not only for the improvement of the performance in predictive tasks, but also quantitatively characterizing the complex relationships between biological entities.

## Methods

### Promoter-centered 3D chromatin interactions

Experimentally validated core promoter-pCRE interaction pair information is required to train a Chromoformer model. In this study, we used publicly available data deposited in the 3DIV database[17] for promoter-centered long-range chromatin interactions compiled by pcHi-C experiments comprehensively conducted for various tissue types. Interactions were characterized at the resolution of HindIII restriction fragments with median and average length of 4797 bp and 5640 bp, respectively. We could also obtain normalized interaction frequencies between the DNA fragments from the database. The obtained frequencies were already processed by a two-step normalization that accounts for the capturability and distance-dependent background signals. Although the significant interactions could be selected based on the estimated FDR values provided along with each interaction, we considered an interaction as significant if the normalized frequency was greater than 1.5 in order to increase the sensitivity of chromatin interactions during the Chromoformer training. Note that a normalized frequency of 1.5 denotes that the ratio between the interaction and background signal is 1.5.

### Training data preparation

Consolidated ChIP-seq read alignments for seven major HMs (H3K4me1, H3K4me3, H3K9me3, H3K27me3, H3K36me3, H3K27ac,

and H3K9ac) were obtained from the Roadmap Epigenomics Project[23] in tagAlign format. This alignment data could make a highly homogeneous training dataset, since the reads were truncated to 36 bp to reduce read length bias originating from the difference in sequencing experiments, and also they were subsampled to a maximum of 30 million reads to homogenize for read depths. After sorting and indexing the alignments with Sambamba v0.6.8[44], read depths along the hg19 reference genome were computed for each base position using *genomecov* command of Bedtools v2.23.0[45]. For each given core promoter or pCREs, we computed log2-transformed binned average signals of seven HMs along non-overlapping genomic windows of size 100 bp, 500 bp, and 2 kbp that fully covers the region and used those values as input features for our model. Since the sizes of pCREs were not fixed in our setting, we zero-padded pCRE feature matrices to make their size agree with that of the core promoter feature matrices. Specifically, we center-aligned the matrix and appended zero-matrices of the appropriate size to the left and right side of the input matrix. To determine prediction target labels, normalized gene expression levels (in RPKM) were also downloaded from the Roadmap Epigenomics Project. RefSeq annotation was used to determine the TSS for each gene. Total 18,955 genes that were appropriately annotated and had expression measurements were selected for model training and evaluation. The whole data collection and preprocessing pipeline is implemented in snakemake workflow management system v6.5.3[46].

Gene-wise prediction targets for the three Chromoformer variants (Chromoformer-clf, Chromoformer-reg and Chromoformer-diff) were derived as follows. Most simply, log2-transformed RPKM values were used for Chromoformer-reg training. For Chromoformer-diff training, we used log2 fold-change of RPKM values between two cell types as prediction targets. To derive binary target labels for Chromoformer-clf training, the median expression values across all genes in each cell type were used as threshold values to assign genes with one of the two labels: highly (1) or lowly expressed (0). In other words, a gene whose expression is above median was assigned with label "1", and the others were assigned with label "0". This formulation of binary classification of gene expression has been widely adopted for various machine-learning approaches for gene expression modeling. Note that these labels ("1" and "0") do not have quantitative meanings, but just denote the ordinal indices of binary classification labels. That is, "0" indicates that the gene is assigned to the first class, and "1" indicates that the gene is assigned to the second class.

## Selection of cell types for model training

For Chromoformer-clf and Chromoformer-reg model training, we only chose a subset of cell types analyzed in the Roadmap Epigenomics Project for which the whole profiles of gene expression, HMs, and 3D chromatin interactions were available. Since the 3D chromatin interaction profiles we used are not the official results of Roadmap Epigenomics but are obtained from an independent source, we manually matched the epigenome IDs (EIDs) and cell type mnemonics from 3DIV database[17]. As a result, the following 11 cell types were selected for Chromoformer training: H1 cells (E003, H1), H1 BMP4 derived mesendoderm (E004, ME), H1 BMP4 derived trophoblast (E005, TB), H1 derived mesenchymal stem cells (E006, MSC), H1 derived neuronal progenitor cultured cells (E007, NPC), HUES64 cells (E016, H1), Liver (E066, LI11), Pancreatic islets (E087, PA), A549 EtOH 0.02pct lung carcinoma (E114, LG), GM12878 lymphoblastoid (E116, GM) and HepG2 hepatocellular carcinoma (E118, LI11).

## Chromoformer model architecture

Chromoformer consists of three modules based on transformer encoders: Embedding, Pairwise Interaction and Regulation transformer.

The embedding transformer has a single encoder layer, which takes a binned average signal matrix $\mathbf{X}_{input}$ of seven HMs at a core promoter and summarizes it into a core promoter embedding matrix $\mathbf{X}_{emb}$ that consists of fixed-sized latent embedding vectors. Before $\mathbf{X}_{input}$ is fed into the module, seven-dimensional input features for each of the $n$ bins are first linearly projected into the dimension of $d_{emb}$ (= 128), then a positional encoding matrix $\mathbf{P}$ of dimension $n \times d_{emb}$ is added to the input feature matrix of the same dimension in an element-wise manner. $\mathbf{P}_{ij}$ is defined as below.

$$\mathbf{P}_{ij} = \begin{cases} \sin\left(\dfrac{i}{10000^{\frac{2k}{d_{emb}}}}\right), & \text{if } j = 2k \\ \cos\left(\dfrac{i}{10000^{\frac{2k}{d_{emb}}}}\right), & \text{if } j = 2k+1 \end{cases} \quad (1)$$

It is worth noting that the inner product between any two-row vectors (e.g., $a$-th and $b$-th rows) in the positional encoding matrix $\mathbf{P}$, namely $\mathbf{P}_a$ and $\mathbf{P}_b$, only depends on the positional distance $|a - b|$ between the two vectors. Therefore, by adding a positional encoding matrix to the input feature matrix, the relative distance between any two features can be recognized in the following multi-head attention layers. A multi-head attention layer in Embedding transformer utilizes a self-attention mechanism to capture inter-dependencies between HM configurations at different positions that contribute to the regulation of gene expression. Importantly, those operations are done separately for multiple heads so that the model can capture different aspects of inter-dependencies between input features. Self-attention operation in the transformer architecture is a special case of scaled dot-product attention where the query, key and value matrices originate from the same sequence of features. Specifically, position-encoded input feature matrix of dimension $n \times d_{emb}$ is linearly projected to produce three matrix $\mathbf{Q}_{emb}$, $\mathbf{K}_{emb}$ and $\mathbf{V}_{emb}$ of dimension $n \times d'_{emb}$, which semantically represents a query, key, and value matrix, respectively. $d'_{emb}$ is set to 64. The $n \times n$ matrix produced by a multiplication of $\mathbf{Q}_{emb}$ and $\mathbf{K}_{emb}^T$ is called a pairwise affinity matrix since each element of the matrix is equivalent to a dot product between the corresponding pair of vectors from $\mathbf{Q}_{emb}$ and $\mathbf{K}_{emb}^T$. It denotes an amount of affinity between the two positions in the input sequence. The pairwise affinity matrix is divided by $\sqrt{d'_{emb}}$ and softmax function is applied to convert self-attention affinities into a weight that sums to 1 for each row. The value matrix $\mathbf{V}_{emb}$ is multiplied with the resulting attention weight matrix to finally produce the output of self-attention operation. The whole process of scaled dot-product can be summarized as below:

$$\text{Attention}_{emb}(\mathbf{Q}_{emb}, \mathbf{K}_{emb}, \mathbf{V}_{emb}) = \text{softmax}\left(\frac{\mathbf{Q}_{emb}\mathbf{K}_{emb}^T}{\sqrt{d'_{emb}}}\right)\mathbf{V}_{emb} \quad (2)$$

In the Embedding transformer, the self-attention operation above is separately done by $m_{emb}$ (= 2) heads and the resulting $m_{emb}$ vectors of dimension $d'_{emb}$ are concatenated to form a single $d'_{emb} \times m_{emb} = d_{emb}$ (= 128) dimensional vector so that the dimension of input feature is preserved. Subsequently, the input sequence of features right before the self-attention is added via residual connection and then they are layer-normalized. The result is then subjected to the linear projection layer into the dimension of $\delta_{emb}$ (= 128), nonlinear activation by rectified linear unit (ReLU) and final linear projection into the dimension of $d_{emb}$. This series of operations involving linear projection, nonlinear activation and another linear projection comprises a position-wise feedforward layer.

$$\text{PositionwiseFeedForward}(x) = \text{Linear}(\text{ReLU}(\text{Linear}(x))) \quad (3)$$

After another residual connection and layer normalization, the core promoter embedding matrix $\mathbf{X}_{emb}$ is finally produced.

Pairwise Interaction transformer consists of two stacked layers to update a core promoter embedding based on its pairwise interaction with each pCRE and produce pairwise interaction embedding matrix $\mathbf{X}_{pair}$. The difference between encoder-decoder attention and self-attention operation in the Embedding transformer is that encoder-decoder attention separately builds query matrix and key-value matrices. Specifically, query matrix $\mathbf{Q}_{pair}$ is derived from $\mathbf{X}_{emb}$ (or $\mathbf{X}_{pair}$ resulting from the first layer), while key and value matrices $\mathbf{K}_{pair}$ and $\mathbf{V}_{pair}$ are built from position-encoded pCRE HM features $\mathbf{X}_{HM}$. In short, the query, key and value matrices for the encoder-decoder attention can be summarized as below, where LinearNoBias denotes a linear projection function without a bias.

$$Q_{pair} = \begin{cases} \text{LinearNoBias}(X_{emb}), & \text{first layer} \\ \text{LinearNoBias}(X_{pair}), & \text{second layer} \end{cases}$$

$$K_{pair} = \text{LinearNoBias}(X_{HM})$$

$$V_{pair} = \text{LinearNoBias}(X_{HM}) \quad (4)$$

Then scaled dot-product attention is conducted between core promoter queries and pCRE key-values as below, where $d'_{pair} = 64$ and $d_{pair} = 128$:

$$\text{Attention}_{pair}(\mathbf{Q}_{pair}, \mathbf{K}_{pair}, \mathbf{V}_{pair}) = \text{softmax}\left(\frac{\mathbf{Q}_{pair}\mathbf{K}_{pair}^T}{\sqrt{d'_{pair}}}\right)\mathbf{V}_{pair} \quad (5)$$

The rest of the operations, including position-wise feedforward layer with $\delta_{pair}$ ( $= 256$), residual connections and layer normalizations are the same as the Embedding transformer, finally producing the pairwise interaction embedding $\mathbf{X}_{pair}$.

To avoid excessive computational load and to make the training batch fit in the memory of a single graphical processing unit (GPU) during training, we only considered at most $i_{max}$ ( $= 8$) pCREs for each core promoter. To determine the set of pCREs participating in the training, all the candidate pCREs were prioritized according to their normalized interaction frequencies with the core promoters since the pCRE that is interacting more frequently is likely to be more informative in predicting the expression of the corresponding gene.

Regulation transformer consists of six stacked layers with gated self-attention mechanism. The key function of the Regulation transformer is to update $\mathbf{X}_{emb}$ along with the whole set of $\mathbf{X}_{pair}$'s at the same time to finally produce the regulatory embedding $\mathbf{X}_{reg}$. To this end, individual embedding vectors that exactly represent the genomic bin where the relevant TSS is located are extracted from $\mathbf{X}_{emb}$ and $\mathbf{X}_{pair}$'s. Then, they are concatenated side by side to form a composite input matrix $\mathbf{X}_{comp}$ of dimension $(i_{max} + 1) \times d_{emb}$ (Recall that $d_{emb} = d_{pair} = 128$). Specifically, those are the vectors at the midpoint of the embedding matrices. Note that for genes having less than $i_{max}$ cis-regulatory interactions, the rest of $\mathbf{X}_{comp}$ was filled with dummy zero vectors. The Regulation transformer does not need a positional encoding since it does not assume any predefined order among the embeddings. We only fix that the first row vector of the composite input matrix is the core promoter embedding. This unordered set of embeddings is fed to a gated self-attention mechanism to allow the model to decide how much it will actively utilize the transformed embedding carrying the interaction information. In addition to the query, key, and value matrices, gated self-attention introduces a gate matrix $\mathbf{G}_{reg}$ that learns the amount of information transfer. The four $(i_{max} + 1) \times d'_{reg}$ matrices used for gated self-attention operation are

computed as below, where $d'_{reg} = 32$:

$$Q_{reg} = \text{LinearNoBias}\left(X_{comp}\right)$$
$$K_{reg} = \text{LinearNoBias}(X_{comp})$$
$$V_{reg} = \text{LinearNoBias}(X_{comp}) \quad (6)$$
$$G_{reg} = \text{LinearNoBias}(X_{comp})$$

Moreover, we added a vector of normalized interaction frequencies $\mathbf{f}$ between the corresponding core promoter-pCRE pair as a bias term to the self-attention matrix to inform the model with the relative affinities of the pairwise interactions. An $(i_{max} + 1) \times (i_{max} + 1)$ bias matrix $\mathbf{B}$ is introduced, whose first row is filled with $\mathbf{f}$ and all the other values are zero. Taken together, the attention operation used in Regulation transformer can be written as below:

$$\text{Attention}_{reg}\left(Q_{reg}, K_{reg}, V_{reg}, G_{reg}\right) = \text{softmax}\left(\frac{Q_{reg}K_{reg}^T}{\sqrt{d'_{reg}}} + \gamma B\right)V_{reg} \circ \sigma(G_{reg}) \quad (7)$$

where $\gamma$, $\sigma$ and $\circ$ represent a learnable scalar coefficient, the sigmoid function and the Hadamard product, respectively.

We concatenated three $\mathbf{X}_{reg}$'s resulting from independent modules learning from 100 bp, 500 bp, and 2 kbp-resolution inputs, respectively. Only the first row of the concatenated matrix, which denotes the cis-regulation-aware embedding vector of the core promoter was extracted and fed into the fully-connected head. In all the three variants of Chromoformer models (Chromoformer-clf, Chromoformer-reg and Chromoformer-diff), the fully-connected head had a single 128-dimension hidden layer with ReLU activation. The fully-connected head for Chromoformer-clf produces a two-dimensional output representing the two prediction logits for each binary expression label, while that for Chromoformer-reg produces a single scalar representing the log2-transformed gene expression value. In Chromoformer-diff, the fully-connected head is fed with a concatenated vector of two multi-scale regulatory embeddings from each cell type and produces a single scalar representing the log2 fold-change of gene expression. Moreover, Chromoformer-diff adopts two auxiliary tasks predicting absolute levels of log2-transformed gene expression for each cell type (Supplementary Fig. 5a). All of the Chromoformer variants were implemented using PyTorch v1.9.0[47].

## Model training and evaluation

All variants of Chromoformer models were trained for 10 epochs with AdamW optimizer[48] and the model resulting from the last epoch was chosen as the final model. The initial learning rate was chosen as $3 \times 10^{-5}$ and was decreased by 13% after each epoch so that it can approximately shrink to half of its value after each of the five epochs. In Chromoformer-clf, cross-entropy between the predicted probability and one-hot encoded binary gene expression label was used as a loss function. In Chromoformer-reg and Chromoformer-diff, mean squared error (MSE) between the predicted scalar and target value was used as a loss function. Batch size was fixed to 64. All the implementations for benchmark deep learning models were obtained from the official code repositories provided by the respective authors. To train the benchmark models, we applied the optimal hyperparameters that were previously identified for each benchmark model.

For GC-MERGE training, we needed to modify our input representations of HM signals and cis-regulatory interactions as per required. ChIP-seq read depths for each 10 kbp bin throughout the genome were calculated using multicov command of Bedtools, and the interaction frequencies between those 10 kbp genomic bins were determined using the pcHi-C experiment results. Since GC-MERGE predictions are made for each of the 10 kbp bins, but not for each gene, ambiguity arises when there are two or more genes in the same bin.

The ambiguity is resolved by choosing a representative gene within each bin and assigning it with the most frequently occurring labels in that bin. This is done at the cost of reduced number of predictable genes. To perform as fair comparison as possible, we went through 4-fold CV for both GC-MERGE and Chromoformer using the same gene set whose expression is predictable by GC-MERGE, by retraining Chromoformer model.

## Analysis of Embedding transformer self-attention
Each Embedding transformer has two independent attention heads, so it produces two corresponding self-attention pairwise affinity matrices $\mathbf{Q}_{emb}\mathbf{K}_{emb}^T/\sqrt{d'_{emb}}$ for each input. Since the full model consists of the three independent single-resolution modules, we can extract six self-attention weight matrices in total. We visualized the softmax-normalized pairwise affinities, i.e., self-attention weights, for Fig. 4. Note that all the self-attention weights were obtained at the time of inference for genes in validation set.

## Computation of the predicted *cis*-regulatory impact, normalization and clustering
To compute predicted cis-regulatory impact (PCRI), we first defined a multi-resolution core promoter embedding as the concatenation of individual core promoter embeddings resulting from the three different input resolutions. Then, PCRI is defined as the Euclidean distance between the multi-resolution core promoter embedding and the multi-resolution regulatory embedding for each gene. Importantly, we standardized each embedding vector before calculating the Euclidean distance to correct for the global shift arising from the transformation itself. Similar to the self-attention analysis, the entire PCRI values discussed in the main text were calculated for genes in each validation set to ensure that the model did not explicitly learn the optimal transformation of latent embeddings reflecting *cis*-regulations for those genes.

To cluster cell types and genes based on PCRI values, the values were standardized (i.e., Z-score normalized) beforehand. The resulting normalized PCRI values in general had bell-shaped distribution around zero (Supplementary Fig. 12). Average linkage hierarchical clustering with correlation similarity was conducted using top 1000 genes having highest across-cell variances of PCRI values.

## Functional enrichment analysis of genes with high PCRI
For each of the four validation folds, we identified top 250 genes with the highest PCRI values separately for each binary label. Functional enrichment analysis of the union of the four gene sets were done using Enrichr[49]. Gene ontology biological process terms with Benjamini-Hochberg adjusted *p*-values <0.05 were selected as significantly enriched terms.

## Definition of polycomb-associated pCREs
To define pCREs associated with Polycomb-bound region, we used irreproducible discovery rate (IDR)-thresholded ChIP-seq peaks for Polycomb subunits determined in H1 cells that are publicly available in ENCODE. Specifically, we downloaded EZH2 and SUZ12 ChIP-seq peaks for PRC2, and RNF2 and CBX8 ChIP-seq peaks for PRC1 (Supplementary Table 1).

## Mouse embryonic stem cell data processing
To evaluate the utility of Chromoformer for species other than human, we processed the ENCODE reference epigenome of ES-Bruce4 mouse embryonic stem cell (mESC) line from its raw histone ChIP-seq reads (Supplementary Table 2). To be consistent with human data, the processing pipeline followed that of Roadmap Epigenomics Project as described below. After downloading FASTQ files, histone ChIP-seq reads were first aligned to mm9 reference genome using bwa v0.7.17-r1188[50]. To normalize the effect of read length, each aligned read was

then truncated up to 36 bp. Also, the read depths were normalized by subsampling the read alignment up to 3 million reads. Processed alignments were converted to genomewide read depth signals using *bedtools genomecov*. Besides, to determine the promoter-pCRE interactions that are used for Chromoformer training, we used the normalized interaction frequencies from publicly available Hi-C interaction matrices of mESC[41]. The bulk RNA-seq gene expression profile for ES-Bruce4 cells was also obtained from ENCODE under file accession ENCFF166EXS [https://www.encodeproject.org/experiments/ENCSR000CGU]. Gencode vM1 gene annotation was used to determine the transcription site and promoter region for each gene.

## CTCF ChIP-seq data processing
To examine the effect of including CTCF signal in model training, we obtained raw CTCF ChIP-seq reads from ENCODE. Only five cell types had available CTCF ChIP-seq data in ENCODE: H1 cells (E003), H1 derived neuronal progenitor cultured cells (E007), A549 lung carcinoma (E114), GM12878 lymphoblastoid (E116), and HepG2 hepatocellular carcinoma cells (E118) (Supplementary Table 3). CTCF ChIP-seq reads were processed in an exact same way as mESC histone ChIP-seq reads (see above), but using hg19 reference genome.

## Reporting summary
Further information on experimental design is available in the Nature Research Reporting Summary linked to this paper.

## Data availability
Histone ChIP-seq read alignments, mRNA expression profiles and chromHMM chromatin states used in this study were downloaded from Roadmap Epigenomics Web Portal[23] [https://egg2.wustl.edu/roadmap/web_portal/index.html]. Normalized interaction frequencies for promoter-centered Hi-C experiments were obtained from hg19 pcHi-C data collection of 3DIV database [http://3div.kr/] under tissue mnemonics H1, ME, TB, MSC, NPC, LI11, PA, LG, and GM. TF ChIP-seq reads targeted for PRC1/2 subunits and CTCF were also downloaded from ENCODE[24] under accession codes specified in Supplementary Tables 1 and 3. TAD and genomic compartmentalization information (in PC1 values) were downloaded from the Supplementary Material of Schmitt et al.[43] Raw histone ChIP-seq reads and the mRNA expression profile for ES-Bruce4 mouse embryonic stem cell data were also downloaded from ENCODE[24] under accession codes specified in Supplementary Table 2. The accession code for the ES-Bruce4 mRNA expression profile was ENCFF166EXS [https://www.encodeproject.org/experiments/ENCSR000CGU]. Hi-C interaction frequency matrices for ES-Bruce4 cells were downloaded from the data repository of Dixon et al.[41] NCBI RefSeq gene annotations were downloaded from UCSC Table Browser [https://genome.ucsc.edu/cgi-bin/hgTables]. Gencode vM1 gene annotations were downloaded from GENCODE [https://www.gencodegenes.org/mouse/release_M1.html]. Source data are provided with this paper.

## Code availability
The source code for Chromoformer model are available at the GitHub repository [https://github.com/dohlee/chromoformer] under doi:10.5281/zenodo.7151966[51]. Pretrained weights for Chromoformer-clf models are available at Figshare under doi:10.6084/m9.figshare.19424807.v1[52]. Code implementations for benchmark models were downloaded from the respective code repositories: DeepChrome [https://github.com/QData/DeepChrome], AttentiveChrome [https://github.com/QData/AttentiveChrome], DeepDiff [https://github.com/QData/DeepDiffChrome], GC-MERGE [https://github.com/rsinghlab/GC-MERGE], and HM-CRNN [https://github.com/pptnz/deeply-learning-regulatory-latent-space].

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

## Acknowledgements

This research was supported by the Collaborative Genome Program for Fostering New Post-Genome Industry of the National Research Foundation (NRF), funded by the Ministry of Science and ICT (MSIT) (NRF-2014M3C9A3063541), Institute of Information & communications Technology Planning & Evaluation (IITP) grant funded by the Korea government (MSIT) [NO.2021-0-01343, Artificial Intelligence Graduate School Program (Seoul National University)], and the Bio & Medical Technology Development Program of the National Research Foundation (NRF) funded by the Ministry of Science & ICT (NRF-2019M3E5D3073375 and NRF-2022M3E5F3085677) (to S. K.).

## Author contributions

D.L., J.Y., and S.K. conceived the experiment(s), D.L. and J.Y. conducted the experiment(s), D.L., J.Y., and S.K. analyzed the results. All authors reviewed the manuscript.

## Competing interests

The authors declare no competing interests.

## Additional information

**Correspondence and requests** for materials should be addressed to Sun Kim.

