## [Peer Review File · Nature Communications]

Learning the histone codes with large genomic windows and three-dimensional chromatin interactions using transformerReviewers' Comments:

Reviewer #1:

Remarks to the Author:

Lee et al., developed a new machine learning model, Chromoformer, to predict gene expression with 3D epigenome information. The model utilized a transformer, a deep-learning algorithm, to reflect the hierarchy of 3D gene regulation in promoter context, promoter-cRE interactions, and a set of 3D pairwise promoter-cRE interactions. As a result, Chromoformer outperforms in terms of prediction of gene expression compared to other existing machine learning methods and also recapitulates well-known gene regulation mechanisms through 'attention' learning. Thus, the proposed method will be highly valuable to elucidate the 3D gene regulation mechanisms. Several points to address my concerns are listed below.

Major comments:

1. The flexibility of Chromoformer: is the current version of Chromoformer applicable to other genome versions, species, and other cell-types?
2. What is the effect of individual histone modification on model performance? The authors selected seven histone modification marks, where five of them were well known core histone modification marks. What is the effect of individual histone modification marks and what is the effect when multiple marks substantially overlap each other? According to the histone code theory, the combination of histone modifications can present various chromatin states. What if they provide chromatin states defined by ChromHMM where the combinatorial histone modification patterns were categorized into various chromatin states?
3. If 3D interaction is critical in Chromoformer, what about the effect of CTCF? As multiple ENCODE cell lines provide CTCF ChIP-seq results, an investigation of whether the inclusion of CTCF affects pairwise interaction and regulation transformer is required.
4. The validation AUC already reached 90-95% in the classification of high and low gene expression patterns for all benchmark methods. Thus, to precisely demonstrate the superior performance of Chromoformer, I suggest to test the prediction accuracy of change of gene expression between cell-types.
5. It is unclear whether the 'attention' in one cell-type is reproducible in another cell-type or cell-type specific.
6. The switch-like suppression at PRC2-bound silencing hubs is interesting observation. However, the conclusion was drawn from only limited number of genes (ex, n=16, n=8, N=13, etc). How could the authors exclude the possibility that such a switch-like contact effect on gene expression is caused by a limited number of genes analyzed in this study? The authors may present that such switch-like contacts to PRC2-bound silencing is associated with cell-type specific gene expression. Further, PRC1 and PRC2 may have different functions according to the cell-types. Have the authors checked PRC1-bound silencing hubs?
7. Compartmentalization of 3D genome (compartment A/B) is another critical parameter that modulates gene expression. Since the presence of active- and inactive-hub is tightly associated with genome compartmentalization, the effect of the inclusion of compartmentalization in the model is worth to further investigate.
8. What if the authors include topologically associating domains, a well-known gene regulatory unit, in the current model? Since all interactions cannot be captured by the pHi-C result, incorporation of TADs may compensate for such experimental limitation.

Reviewer #2:

Remarks to the Author:

In this paper, Kim and colleagues present Chromoformer, a transformer-based deep learning model that predicts gene expression levels based on histone modifications at promoters as well as 3D genome-interacting enhancers. Transformer is an important new method, and the application of transformer to gene expression prediction is novel. The authors compare the chromoformer with previous methods.

1. In the comparison with previous methods, the so-called outperformance is not very high. The existing methods appear to perform with approximately 92% performance, whereas Chromoformer's performance is around 93-94%. This is not a big difference, and I would hesitate to say that Chromoformer "outperforms" on the basis of this difference.
2. In line with my comment 1, separation of gene expression data into expressed and not expressed based on median gene expression levels has been done before and we know that existing gene expression models can do this job. However, gene expression is a very wide range, and I do not think most existing gene expression models can predict gene expression in a very quantitative manner. Can Chromoformer do this? If so, this will give more evidence that Chromoformer indeed can perform better in terms of gene expression prediction.
3. While the use of transformer methods to predict gene expression is useful, I feel that the novelty of the biologically significant findings is not very high. We know about transcription factories and silencing hubs already. Does this prediction method make any novel predictions that generate novel testable biological hypotheses that can then be tested by the authors?
4. Previous methods investigating 3D genome organization such as TargetFinder have encountered issues (discussed in Xi and Beer, PLoS Computational Biology, 2018 and Cao and Fullwood, Nature Genetics, 2019). As Xi and Beer say "We report an experimental design issue in recent machine learning formulations of the enhancer-promoter interaction problem arising from the fact that many enhancer-promoter pairs share features. Cross-fold validation schemes which do not correctly separate these feature sharing enhancer-promoter pairs into one test set report high accuracy, which is actually arising from high training set accuracy and a failure to properly evaluate generalization performance." Have the authors evaluated their method to see if this issue is present, or not a problem?

Minor comments

1. Figure 2 – "CV" is not explained in the figure legend
2. I am still not very clear about how the training dataset was prepared. The authors say "For each cell type, the median expression values across all genes were used as threshold values to assign genes with one of the two labels: highly (1) or lowly expressed (0)." – does this mean that if the gene expression was above the median expression, then it was denoted as "1"?

Reviewer #3:

Remarks to the Author:

In this article, the authors present a transformer-based, three-dimensional (3D) chromatin conformation-aware deep learning architecture named Chromoformer that achieves the state-of-the-art performance in the quantitative deciphering of the histone codes in gene regulation. The authors decomposed the architecture into three phrases understand the complex dynamics of cis-regulations involving multiple layers (1) cis-regulation by core promoters, (2) 3D pairwise interaction between a core promoter and a putative cis-regulatory regions (pCREs) and (3) a collective regulatory effect imposed by the set of 3D pairwise interactions.

Here are my reviews for this article

1. The authors developed three transformer models(Embedding transformer, pairwise transformer and Regulation Transformer). For each transformer model, the input features are core promoter features and pCRE features. It is unclear how these features were generated. Did any deep learning models were used to generate these features? Some details are necessary to understand this step.
2. The architecture of "Embedding Transformer" and "Pairwise Interaction Transformer" is same. Is there any specific reason to give different name? It could be misleading that you have developed two different transformer architecture.
3. One of the main key points of the regulation transformer is to "normalized interaction frequencies between the corresponding core promoter-pCRE pair as a bias term to the self-attention matrix to inform the model with the relative affinities of the pairwise interactions". What is the reason of using "normalized interaction frequencies"? How were normalized interaction frequencies derived?
4. The authors used three modules (100bp, 500bp and 2000bp modules) and the outputs concatenated before final prediction using fully connected layers. Can self-attention based concatenation be applied so that the multi-scale regulatory embedding is generated with their attention score? As in the normal concatenation, are you giving equal weightage to all three outputs?
5. Also, why the authors specifically consider 100bp, 500bp and 2000bp input regions? Did the authors check the prediction result only for two regions like 100bp and 500bp OR 500bp and 2000bp?

Response to reviewer's comments

We would like to thank all the reviewers for the insightful and constructive suggestions. According to the comments, we tried to improve the quality of the manuscript through additional experiments and further discussions on model performances and biological findings. Please see below our detailed response to reviewers' comments in blue.

Reviewer #1 (Remarks to the Author):

Lee et al., developed a new machine learning model, Chromoformer, to predict gene expression with 3D epigenome information. The model utilized a transformer, a deep-learning algorithm, to reflect the hierarchy of 3D gene regulation in promoter context, promoter-cRE interactions, and a set of 3D pairwise promoter-cRE interactions. As a result, Chromoformer outperforms in terms of prediction of gene expression compared to other existing machine learning methods and also recapitulates well-known gene regulation mechanisms through 'attention' learning. Thus, the proposed method will be highly valuable to elucidate the 3D gene regulation mechanisms. Several points to address my concerns are listed below.

Major comments:

1. The flexibility of Chromoformer: is the current version of Chromoformer applicable to other genome versions, species, and other cell-types?

We appreciate this valuable comment. We agree that the flexibility and wide applicability of a machine learning model in computational biology applications are of great importance to facilitate further research derived from the computational model. To this end, we carried out two additional (1) inter-species and (2) inter-cell-type validation experiments to support the flexibility of Chromoformer model training.

First, we conducted an experiment to show that Chromoformer can be trained or evaluated on the epigenomes of species other than the human. To this end, we processed the ENCODE reference epigenome of ES-Bruce4 mouse embryonic stem cell (mESC) line from its raw sequencing reads. Also, we used the normalized interaction frequencies from publicly available Hi-C interaction matrices of mESC (Dixon et al., 2012) to determine the promoter-pCRE interactions that are used for Chromoformer training.

As a result, we observed that Chromoformer achieves high validation AUC for mESC data (Average AUC of 95.30, 4-fold cross validation, **Figure R1-1a**), which is on par with the performances from models trained with human embryonic stem cell line H1 and HUES64 (Average AUC of 93.52 and 94.28, respectively, 4-fold cross validation, **Figure R1-1b**). This result shows that the Chromoformer model can also be trained with epigenomic data from different species, and also with different versions of the reference genome. Having confirmed the applicability of Chromoformer to the mouse epigenome, we were curious to what extent is the learned grammar of the regulatory histone codes shared across the human and mouse epigenomes. To this end, we evaluated the performances of Chromoformer models trained on H1 or HUES64 for the prediction of mESC gene expression and vice versa. We observed that the cross-species performances were reasonable for hESC-to-mESC predictions (AUC of 93.87 and 93.42 from models trained with H1 and HUES64 epigenomes,

Figure R1-1 (Related to Supplementary Fig. 17 and 18). Inter-species and inter-cell type prediction performances of Chromoformer-clf models. Performances of Chromoformer-clf models, trained with ES-Bruce4 mouse embryonic stem cells (mESC) or human embryonic stem cells (hESCs), are shown for the prediction of (a) mESC gene expression and (b) hESC gene expression. Log₂-transformed histone modification signals surrounding (c) human SOX2 TSS and (d) mouse Sox2 TSS. (e) Cross-cell type gene expression prediction performances. Colors represent the relative validation AUC compared to the matched cell type Chromoformer-clf model (i.e., trained and evaluated for the same cell type).

respectively, **Figure R1-1a**) as well as mESC-to-hESC predictions (AUC of 92.24 and 92.90 for H1 and HUES64 gene expressions, respectively, **Figure R1-1b**). These results suggest that the roles of individual histone mark in gene regulation is not only qualitatively conserved between mouse and human, but the link between the quantity of histone modifications and the amount of gene expression is also conserved to a reasonable extent, as exemplified in **Figure R1-1c and d** showing the quantitative similarities of histone marks for human SOX2 and mouse Sox2 gene expressed to a similar extent in both species.

Next, to examine the inter-predictability of Chromoformer across human cell types, we evaluated the performance of a Chromoformer model trained for one cell type on the other cell types. For every cell type examined, the best performance was achieved by the model directly trained for that cell type (matched cell type model, diagonal values in **Figure R1-1e**). However, we observed that a Chromoformer model trained in one cell type was still applicable to other cell types to some degree (relative validation AUC >92%, off-diagonal values in **Figure R1-1e**). We also found that the tendency of cross-cell type prediction performances was concordant with the similarity of cell types. For example, two hESC lines (H1 and HUES64) showed highly similar performance tendencies, and cancer (HepG2 and A549) and immortalized (GM12878) cell lines were clustered together (**Figure R1-1e**). Altogether, these results suggest that Chromoformer trained in cell type-specific manner indeed learns cell type-specific features of gene regulation, but still captures the general rules that can be commonly applied to other cell types. Thanks to the reviewer's comment, we added the above results to the Discussion (Supplementary Fig. 17 and 18) of the revised manuscript by showing the

Figure R1-2 (Related to Supplementary Fig. 10). Histone mark ablation study. (a) Bars show the decrease in validation AUC when each histone mark was excluded from Chromoformer-clf training. Performance decreases were averaged across all the 11 cell types. Error bars denote standard error of the mean. (b) Correlation between pairs of histone mark signals. (c) Each row in the left panel shows the combination of features that were ablated simultaneously, and the corresponding row in the right panel show the decrease in AUC. The black box highlights the independent impact of H3K36me3 ablation. (d) Emission probabilities of the seven histone marks for each of the 50 chromatin states inferred from chromHMM model. Similar pair of histone mark combinations were matched between panels (c) and (d).

expandability of the framework.

Meanwhile, to enhance the applicability of Chromoformer, we developed and made public an automated pipeline for the preparation of the input data for Chromoformer. Given the raw histone ChIP-seq read or alignment files, the reference genome, gene annotations and the catalog of 3D chromatin interactions, the pipeline aligns the read to the genome, computes the read depths and prepares the input matrices representing promoter/pCRE histone signals as binary files so that it can be readily fed to the Chromoformer model while training and evaluation. For the implementation of the pipeline, please refer to the github repository dedicated for this study (<https://github.com/dohlee/chromofomer>).

2. What is the effect of individual histone modification on model performance? The authors selected seven histone modification marks, where five of them were well known core histone modification marks. What is the effect of individual histone modification marks and what is the effect when multiple marks substantially overlap each other? According to the histone code theory, the combination of histone modifications can present various chromatin states. What if they provide chromatin states

Figure R1-3 (Related to Supplementary Fig. 19). Incorporating CTCF binding signals in Chromoformer training. (a) Performances of Chromoformer-clf models trained with or without CTCF binding signals. (b) Performances of Embedding transformer-only Chromoformer-clf models trained with or without CTCF binding signals. Values in parentheses denote the amount of performance improvement when CTCF binding signals were used. (c) Average CTCF ChIP-seq read depths around TSS. Read depth signals were grouped and averaged based on the binary gene expression states.

defined by ChromHMM where the combinatorial histone modification patterns were categorized into various chromatin states?

We are grateful for this suggestion. According to the suggestion, we first conducted feature ablation study to see the effect of individual histone marks on the model performance. We excluded one histone marks for each model training episode, and measured the decrease in performance compared to the original model trained with all the seven histone marks. By doing this, we could determine the contribution of each histone marks and their combinations in modeling gene regulation.

The results of single-feature ablation experiments are shown in **Figure R1-2a**, and we found that the ablation of H3K36me3 mark led to the most dramatic decrease in performance. This can be explained by the observation that H3K36me3 level was the least correlated with other six marks (**Figure R1-2b**), thus its ablation can hardly be compensated by the remaining marks. Next, to examine the combinatorial effect of histone marks on gene expression prediction, we simultaneously excluded two or more histone marks and measured the amount of performance degradation similarly. We observed that a majority of histone combinations leading to the greatest decrease in model performance qualitatively coincided with the combination characterizing the chromatin states previously determined by chromHMM (**Figure R1-2c-d**). Notably, the detrimental effect of H3K36me3 ablation seemed almost independent of those combinations, again reflecting its unique spatial distribution. This suggests that Chromoformer internally captures the well-known combinations of histone modifications for chromatin states. We added these results to the revised manuscript and strengthened the discussion on H3K36me3 mark.

3. If 3D interaction is critical in Chromoformer, what about the effect of CTCF? As multiple ENCODE cell lines provide CTCF ChIP-seq results, an investigation of whether the inclusion of CTCF affects pairwise interaction and regulation transformer is required.

We appreciate this insightful comment. According to the suggestion, we carried out experiments by training Chromoformer models with CTCF ChIP-seq signals in addition to the seven histone ChIP-seq signals. Experiments were done using the following cell types for which the raw CTCF ChIP-seq data were available in ENCODE: H1 (E003), H1 derived neuronal progenitor cultured cells (E007), A549 EtOH 0.02pct lung carcinoma (E114), GM12878 lymphoblastoid (E116) and HepG2 hepatocellular carcinoma (E118).

Adding CTCF binding information resulted in moderate performance improvements in four out of five cell types examined (average AUC increase of 0.06~0.11 for cell types E003, E007, E114 and E116), although the improvement did not reach statistical significance (**Figure R1-3a**). In addition, following the reviewer's suggestion, we examined whether the incorporation of CTCF binding information will compensate the absence of pairwise interaction and regulation transformers in Chromoformer training by indirectly providing models with the information about 3D chromatin interactions. To this end, we measured the performance of an 'Embedding transformer-only' Chromoformer variant when trained with CTCF ChIP-seq signals. As a result, we observed much greater performance increase (average AUC increase of 0.12~0.14 for cell types E003, E007, E114 and E116, **Figure R1-3b**), but the performances did not reach that of intact Chromoformer models trained without CTCF signals.

Taken together, the increased performance, if not dramatic, suggest that incorporating CTCF binding signals at promoters (or 40kbp window around TSS) can provide some information that may not be explained by histone modification states. This is consistent with the recent discovery highlighting the role of promoter-proximal CTCF binding in gene activation through distal enhancer-promoter interaction (Naoki et al., 2021). **Figure R1-3c** supports that highly expressed genes (i.e., genes with binary label "1") show prevalent CTCF binding near transcriptional start sites compared to lowly expressed genes (i.e., genes with binary label "0"). Thus, these results show the discriminative power of CTCF binding signals in predicting gene expression states. Meanwhile, our observation that the 'Embedding transformer-only' Chromoformer trained with CTCF signals did not reach the performance of intact Chromoformer model demonstrates that we still need another transformers modeling promoter-pCRE interactions and their collective regulatory effects. We conjecture that the configurations of histone modification at distal pCREs deliver richer regulatory information compared to CTCF signal alone, which indicates E-P contacts through chromatin folding near promoters.

Since we decided to confine our results in the main text to histone modifications, we added these results on CTCF in the Discussion of the revised manuscript (Supplementary Fig. 19), by emphasizing the expandability of Chromoformer models to incorporate any genomic signal tracks that may account for the unexplained portion of gene regulation, such as TF ChIP-seq signals.

4. The validation AUC already reached 90-95% in the classification of high and low gene expression patterns for all benchmark methods. Thus, to precisely demonstrate the superior performance of Chromoformer, I suggest to test the prediction accuracy of change of gene expression between cell-types.

We would like to thank the reviewer for this suggestion. We agree that it is necessary to additionally demonstrate the performance of Chromoformer in additional tasks beyond a binary gene expression prediction task. Especially, predicting the relative change of gene expression between cell types based on histone modification states will be more challenging for computational models, because the model needs to precisely learn how to quantitatively translate the difference of histone codes into relative difference of gene expression activity.

According to the comment, we newly designed and trained a variant of Chromoformer that predicts

Figure R1-4 (Related to Fig. 2 and Supplementary Fig. 5). Chromoformer-diff model architecture and performance. (a) Schematic illustration of Chromoformer-diff model architecture. (b) Performances of Chromoformer-diff models and other benchmark models in predicting gene expression fold-change between two cell types. (c) Pairwise performance comparison between DeepDiff and Chromoformer-diff. (d) Examples of Chromoformer-diff predictions for \log_2 (expression fold change). Note that 4-fold cross-validation predictions were pooled into a single plot.

the change of gene expression between cell-types. We called the model 'Chromoformer-diff' to differentiate it from the Chromoformer model for binary gene expression classification (which we call

'Chromofomer-clf' in the revised manuscript). Importantly, this deep learning-based prediction of differential expression was first addressed by DeepDiff (Sekhon et al., 2018), and thus we benchmarked the performance of our Chromofomer-diff model with that of DeepDiff.

The architecture of the Chromofomer-diff model is illustrated in **Figure R1-4a**. Given relevant histone modification contexts for a target gene in two difference cell types, a model was trained to predict log2 fold-change of gene expression between the two cell types using two Chromofomer backbones with shared weights. We added the detailed description of Chromofomer-diff model architecture in the Results and Supplementary Fig. 5 of the revised manuscript. We trained Chromofomer-diff with seven cell type-pairs that have been already validated in the original DeepDiff paper (Sekhon et al., 2018). We also measured the predictive power of the ratio of the prediction probabilities obtained from classifier models (Chromofomer-clf and benchmark classifiers) in predicting the log2 expression fold change of genes (i.e., The correlation between \log_2 (predicted probability of gene i in cell type A / predicted probability of gene i in cell B) and \log_2 (expression fold change)). As a result, we observed that Chromofomer-diff outperformed DeepDiff and the other classifier models (**Figure R1-4b and c**). The across-cell type average Pearson's correlation coefficient between the predicted and true expression fold change increased from 0.577 to 0.635, when DeepDiff and Chromofomer-diff was compared. Qualitatively, we found that Chromofomer-diff tended to show better predictions for genes with extreme differences between the two cell types (**Figure R1-4d**).

Meanwhile, to demonstrate the quantitative modeling performance of Chromofomer model similarly, we also trained a variant of Chromofomer model to directly predict the gene expression level for individual cell types according to the query of reviewer #2. In other words, we let Chromofomer solve gene expression prediction as a regression task. For the results of regression-based Chromofomer, please refer to our response of question #2 raised by reviewer #2 and **Figure R2-2**.

5. It is unclear whether the 'attention' in one cell-type is reproducible in another cell-type or cell-type specific.

We agree that it is necessary to show whether the findings from the learned self-attention weights of Embedding transformers are consistent across different cell types. According to the reviewer's suggestion, we inspected the learned attention weights from each Chromofomer model trained for different cell types. As a result, we found that the tendency of Embedding transformers to attend to 4-6kbp downstream regions from TSS is highly consistent throughout cell types (**Figure R1-5**), which suggest that the transcriptional elongation signal at the gene body consistently gave additional information to the model about the transcriptional state of genes regardless of cell type. We added these results in the revised manuscript (Supplementary Fig. 9).

6. The switch-like suppression at PRC2-bound silencing hubs is interesting observation. However, the conclusion was drawn from only limited number of genes (ex, n=16, n=8, N=13, etc). How could the authors exclude the possibility that such a switch-like contact effect on gene expression is caused by a limited number of genes analyzed in this study? The authors may present that such switch-like contacts to PRC2-bound silencing is associated with cell-type specific gene expression. Further, PRC1 and PRC2 may have different functions according to the cell-types. Have the authors checked PRC1-bound silencing hubs?

We would like to thank the reviewer for the intriguing comment. According to the comment, we further

Figure R1-5 (Related to Supplementary Fig. 9). Across-cell type consistency of self-attention weights learned by the Embedding transformer of Chromoformer-clf. Epigenome ID denoting the corresponding cell type is shown above each plot.

assessed whether the switch-like behavior of PRC2-bound silencing hubs has true biological significance or was an experimental artifact due to the small number of genes.

To this end, we first examined whether we could rescue more genes that are affected by switch-like suppression of PRC2 by introducing additional ChIP-seq data targeting PRC2 subunits. We could obtain several replicates of EZH2 and SUZ12 ChIP-seq peaks from ENCODE, and we observed that using merged peaks rescued more genes compared to the previous analyses.

To provide additional support for the switch-like suppression of PRC2 observed in this study, we investigated whether it was a common phenomenon across cell types or not. As the suppression of developmental genes by PRC2 is predominantly observed and studied for stem cell types (Chan and Morey, 2019), we expected that the sigmoidal patterns would only be exhibited in undifferentiated cell types (E003~E016). As expected, we observed such patterns for H1 embryonic stem cell (E003), mesendoderm (E004), neural progenitor cells (E007), and HUES64 embryonic stem cell (E016) for both EZH2 and SUZ12 binding site analyses. However, we could not observe similar results in trophoblast (E005) and mesenchymal stem cells (E006). Interestingly, this result is highly consistent with the previously demonstrated cell type hierarchy based on H3K27me3 signal in Polycomb-associated repressive domains (Kundaje et al., 2015), in which H1 cells (E003), mesendoderm cells (E004), neural progenitor cells (E007) and HUES64 cells (E016) were clustered together. This suggests the similarities of the epigenetic regulation dynamics among those specific cell types were

Figure R1-6 (Related to Supplementary Fig. 14, 15 and 16). Tendency of PCRI values depending on pCREs harboring PRC2 or PRC1-binding sites. Relationships between predicted *cis*-regulatory impact (PCRI) and the number of putative *cis*-regulatory elements (pCREs) with (a) EZH2 and (b) SUZ12 binding, which are subunits of polycomb repressive complex 2 (PRC2), are shown. Similarly, panel (c) and (d) shows the association between PCRI and the number of pCREs having RNF2 and CBX8 binding sites, respectively, which are subunits of PRC1.

recapitulated by the interpretation of trained Chromoformer models.

Finally, according to the reviewer's comment, we conducted similar analyses for PRC1-bound silencing hubs. To locate PRC1 throughout the genome, we utilized ChIP-seq peaks for RNF2 and CBX8 subunits in ENCODE and observed similar trends of sigmoidal increase of PCRI values following the increasing number of pCREs with PRC1-binding sites in H1, mesendoderm, neural progenitor and HUES64 cells (**Figure R1-6c and d**). The marked similarity in the characteristic of

Figure R1-7 (Related to Supplementary Fig. 20). Incorporating genomic compartment states in Chromoformer training. (a) Performances of Chromoformer-clf models trained with the first principal component (PC1) values of the correlation matrix made with Hi-C contact matrix. (b) Correlation between gene expression and the PC1 value. (c) Distribution of gene expression based on the compartment A/B state.

PRC1 and PRC2 may be due to the interdependence of PRC1 and PRC2 for their binding to Polycomb response elements (Kahn et al., 2016) in those cell types. Thanks to the reviewer's comment, we have improved the discussion on the switch-like suppression of Polycomb-bound bodies by adding the results above to the revised manuscript (Supplementary Fig. 14, 15 and 16).

7. Compartmentalization of 3D genome (compartment A/B) is another critical parameter that modulates gene expression. Since the presence of active- and inactive-hub is tightly associated with genome compartmentalization, the effect of the inclusion of compartmentalization in the model is worth to further investigate.

Figure R1-8 (Related to Supplementary Fig. 8). Contribution of inter- and intra-TAD chromatin interactions in Chromoformer training. (a) Schematic illustration of inter- and intra-TAD chromatin interactions. (b) Proportion of inter-TAD interactions used in Chromoformer training for each cell type. (c) Performance differences of Chromoformer-clf models when inter- and intra-TAD interactions were excluded from training. Average validation AUC scores were separately measured for genes with and without at least one *cis*-regulatory interaction.

We agree with the reviewer's comment in that the compartmentalization of 3D genome beyond one-to-one contacts between genomic fragments may provide additional information on the regulation of gene expression. To this end, we utilized the 1Mbp-resolution compartment A/B calls from Schmitt et al., 2016, which covers all the cell types used in this study.

To inform the model with compartment A/B status, we added PC1 values used in compartment calling as input features along with histone ChIP-seq signals and re-trained Chromoformer models. Unexpectedly, we observed that adding PC1 values did not give significant overall performance gain (**Figure R1-7a**). To examine the reason for this result, we tested for the association between 3D genome compartmentalization and gene expression values. As is widely established, we could observe that the state of 3D genome compartmentalization showed significant association with the expression level of gene in our data (**Figure R1-7b-c**), but the absolute level of the association (Pearson's correlation coefficient 0.11~0.19) was not high enough to have reasonable predictability for gene expression levels. Thus, we concluded that the predictive power of compartment-level features did not exceed that of gene-level histone modification features. We added the results to the Discussion of the revised manuscript (Supplementary Fig. 20).

8. What if the authors include topologically associating domains, a well-known gene regulatory unit, in the current model? Since all interactions cannot be captured by the pHi-C result, incorporation of TADs may compensate for such experimental limitation.

We thank the reviewer's insightful comment. According to the suggestion, we investigated whether the inclusion of TAD information would improve model performance. We initially thought that TAD information may be complementarily refine pHi-C interaction data in two ways (**Figure R1-8a**): (1)

by rescuing false-negative interactions as the reviewer's comment has pointed out, and (2) by reducing false-positive interactions, which will mainly include inter-TAD interactions. Regarding the former case (rescuing false-negative interactions), we agree that using TAD dramatically reduces the search space of plausible promoter-pCRE interactions and helps imputing the missing interactions in the pHi-C data, but we thought that exploiting it directly to Chromoformer training will be hard since accurate *de novo* prediction of promoter-pCRE interactions within a TAD is still a difficult task, although there exist several deep learning approaches (Singh et al., 2019; Avsec et al., 2021). Therefore, we focused our experiments on the latter case (i.e., reducing false-positive interactions).

We obtained 40kb-resolution TAD annotation for 11 cell types from Schmitt et al., 2016. Using the TAD information, we classified each promoter-pCRE interaction used for training as either of intra-TAD or inter-TAD interaction and found about 16%~22% of interactions were inter-TAD interactions (**Figure R1-8b**). Indeed, not all of those inter-TAD interactions would be biologically irrelevant, but to see the effect of using a set of 3D chromatin interactions with increased signal-to-noise ratio, we just discarded inter-TAD interactions from Chromoformer-clf training. As a control, we also examined the effect of excluding intra-TAD interactions. Resultingly, we observed that the exclusion of inter-TAD interactions neither consistently increased nor decreased the model performance (**Figure R1-8c**), suggesting that inter-TAD interactions were already having negligible or dispensable effect to the model performance. On the other hand, excluding intra-TAD interactions resulted in marginal performance degradation in several cell types only for genes having *cis*-regulatory interactions, supporting the relative importance of intra-TAD interactions compared to inter-TAD interactions in Chromoformer-clf training (**Figure R1-8c**). Meanwhile, such marginal performance decreases imply there still exist meaningful regulatory information in inter-TAD interactions.

We updated our manuscript to include these results on the contribution of inter- and intra-TAD interactions in Chromoformer training (Supplementary Fig. 8).

Reviewer #2 (Remarks to the Author):

In this paper, Kim and colleagues present Chromoformer, a transformer-based deep learning model that predicts gene expression levels based on histone modifications at promoters as well as 3D genome-interacting enhancers. Transformer is an important new method, and the application of transformer to gene expression prediction is novel. The authors compare the chromoformer with previous methods.

1. In the comparison with previous methods, the so-called outperformance is not very high. The existing methods appear to perform with approximately 92% performance, whereas Chromoformer's performance is around 93-94%. This is not a big difference, and I would hesitate to say that Chromoformer "outperforms" on the basis of this difference.

We appreciate the reviewer for pointing this issue out. We agree that the existing models exhibit good performances (~92 of AUC) for the binary gene expression classification task. Thus, to show the outperformance of Chromoformer in more challenging tasks on gene expression prediction, we carried out two additional benchmark experiments.

1. Predicting the values of expression levels as a regression task (Chromofomer-reg)

To examine the capacity of the Chromofomer model for quantitative modeling of gene expression regulation, we formulated a regression task to predict log₂-transformed expression levels of genes and trained Chromofomer and benchmark models. As a result, Chromofomer-reg showed 0.667 of R² value between true and predicted expression values, while the best competitor, DeepChrome, showed 0.627. For a more detailed description of Chromofomer-reg model and its evaluation results, please refer to our response for question #2 of reviewer #2 and **Figure R2-2**.

2. Predicting the fold-change of expression levels between two cell types (Chromofomer-diff)

Next, a variant of Chromofomer is trained to predict the fold-change of a gene based on histone modification configurations in two different cell types. This formulation of gene expression prediction is more challenging since the model should learn to map histone contexts of two different cells into a unified embedding space and at the same time learn how to quantitatively translate the discrepancy of embeddings into the relative difference of gene expression levels. As a result, the average Pearson's correlation coefficient between the predicted and true expression fold-change was 0.635, while that of state-of-the-art model, DeepDiff, was 0.577. For a more detailed description of Chromofomer-diff model, please refer to our response for question #4 of reviewer #1 and **Figure R1-4**.

Altogether, the consistent performance increases of Chromofomer models in various gene expression prediction tasks strongly underscore the contribution of additional biological features, namely, 3D chromatin interactions. In other words, it implies that certain portion of regulatory information can only be effectively explained by information embedded in *cis*-regulatory elements that is conveyed to promoters through 3D chromatin interactions, but not solely by core promoter histone codes. We think that suggesting a new deep learning model architecture that effectively extracts such information is one of the important technical novelties of this study. Thanks to the reviewer's comment, we revised the manuscript accordingly to improve the discussion on performance improvement.

2. In line with my comment 1, separation of gene expression data into expressed and not expressed based on median gene expression levels has been done before and we know that existing gene expression models can do this job. However, gene expression is a very wide range, and I do not think most existing gene expression models can predict gene expression in a very quantitative manner. Can Chromofomer do this? If so, this will give more evidence that Chromofomer indeed can perform better in terms of gene expression prediction.

We agree that since gene expression values are continuous and their range varies several orders of magnitude in general, the binary classification formulation of gene expression prediction task may not be sufficient to support the quantitative predictive power of Chromofomer. To further support the quantitative modeling power of Chromofomer, we designed a modified version of Chromofomer as a regression model that predicts the exact value of gene expression instead of binary labels (Chromofomer-reg model, **Figure R2-2a**). To this end, we changed the last fully-connected layers of the Chromofomer to predict a single scalar value instead of binary logits. Then, Chromofomer-reg models were trained from scratch to predict log₂-transformed expression value. All the training configurations were kept the same except that mean squared error was used as a loss function. By modifying benchmark models into corresponding regression models in a similar way, we could demonstrate that Chromofomer performs better not only in classification task but also in regression

Figure R2-2 (Related to Fig. 2 and Supplementary Fig. 7). Chromoformer-reg model architecture and performances. (a) Schematic illustration of Chromoformer-reg model architecture. (b) Performances of Chromoformer-reg models in terms of Pearson correlation coefficient. (c) Performances of Chromoformer-reg models in terms of R^2 value.

task compared to benchmark models. Specifically, Chromoformer-reg showed higher Pearson's correlation coefficient and R^2 score (Figure R2-2b and c). We added these results in Results and Supplementary Fig. 7 of the revised manuscript to support the ability of Chromoformer to predict gene expression in a quantitative manner.

3. While the use of transformer methods to predict gene expression is useful, I feel that the novelty of the biologically significant findings is not very high. We know about transcription factories and silencing hubs already. Does this prediction method make any novel predictions that generate novel testable biological hypotheses that can then be tested by the authors?

We would like to thank the reviewer for the valuable suggestion. We agree that the existence of transcription factories and silencing hubs inside the nucleus has already long been established by existing studies. To go beyond just demonstrating the existence of those intranuclear hubs, we thought that the novelty of this study can be found in that it provides a systematic way to infer the collective

Figure R2-3 (Related to Fig. 6). Differential *cis*-regulome analysis using PCRI values. (a) Hierarchical clustering of top 1,000 genes having highest normalized PCRI variances across cell types. Representative GO terms enriched for the corresponding set of genes are shown on the right. (b) Histone modification landscape around the transcription start site of *CCN2* and its pCREs in Liver (E066) and HepG2 (E118) cells. Red shades denote promoter regions and blue shades denote pCRE regions interacting with the promoter. Red arrow represents a putative enhancer region that seems to be only active in HepG2 cells.

effect of *cis*-regulatory interactions in highly quantitative manner. This can be achieved through the proposed value named predicted *cis*-regulatory impact (PCRI). In this regard as well as according to the reviewer's suggestion, we conducted additional analyses to provide further biological findings based on PCRI values.

We thought that the whole collection of PCRI values for each cell type can be considered as a gene-centric representation of the *cis*-regulome. That is, the *cis*-regulation functioning in each cell type can be summarized as a ~20,000-dimensional vector having PCRIs as its elements. We were first curious whether such representation of *cis*-regulome can reveal similarities and differences between cell types. To this end, we selected 1,000 genes whose PCRIs were highly variable across cell types and performed hierarchical clustering based on their PCRI values. As a result, genes associated with cell type-specific functions had high PCRI values and were clustered together (**Figure R2-3a**), implying the coordinated *cis*-regulation imposed on the genes.

Meanwhile, we discovered a small subset of genes that had high PCRIs in HepG2 hepatocellular carcinoma (HCC) cells, but not in healthy adult liver tissue (**Figure R2-3a** **black box**). Interestingly, we could not find any biological terms significantly enriched for those genes unlike other clustered gene sets in the analysis. We speculated that they represent a consequence of cancer-specific aberrant *cis*-regulation occurring in a stochastic manner. Although the genes did not show collective functional enrichment, we could identify four individual genes (*GNA12*, *TRIB3*, *CCN2* and *RBM39*) tightly implicated in HCC, which can be thought as epigenomic "hits" by aberrant *cis*-regulation. In accordance with the tendency of PCRIs, the expression of the four genes were 9.3-, 6.0-, 4.1-, and 3.7-fold higher in HCC than in healthy liver cells, respectively.

We further tried to interpret why did the Chromoformer predict high PCRI values for those genes by visualizing the histone modification landscape surrounding genes. For example, **Figure R2-3b** shows histone modification landscapes around *CCN2* in healthy liver and HCC cells. Comparing two landscapes revealed a putative enhancer region that is only active in HCC (**Figure R2-3b** **red arrow**), which may explain higher PCRI as well as higher expression of *CCN2* in HCC. It highlights that the in-depth interpretation of Chromoformer model prediction in the form of differential *cis*-regulome analysis can reveal an epigenomic origin of malignant gene expression. As the histone modification profiles for more cell types become available, we expect that this data-driven approach will be more effective in revealing cancer-specific or condition-specific *cis*-regulation events on the basis of the promoter-pCRE interactions. Thanks to the reviewer's comment, we improved the results on PCRI analysis in the revised manuscript (Figure 6).

Besides, according to the comment of Reviewer 1, we also added further discussion on the silencing hubs through across-cell type analysis and PRC1-bound hubs. For the results of the analyses, please refer to our response of question #6 raised by reviewer #1 and **Figure R1-6**.

4. Previous methods investigating 3D genome organization such as TargetFinder have encountered issues (discussed in Xi and Beer, PLoS Computational Biology, 2018 and Cao and Fullwood, Nature Genetics, 2019). As Xi and Beer say "We report an experimental design issue in recent machine learning formulations of the enhancer-promoter interaction problem arising from the fact that many enhancer-promoter pairs share features. Cross-fold validation schemes which do not correctly separate these feature sharing enhancer-promoter pairs into one test set report high accuracy, which is actually arising from high training set accuracy and a failure to properly evaluate generalization performance." Have the authors evaluated their method to see if this issue is present, or not a problem?

We thank the reviewer for pointing out this critical issue with useful references. We indeed agree that

the evaluation of machine learning (ML) models based on genomic features should be done carefully, since promoters in training and validation data can share common cis-regulatory elements if we just split genes randomly, and thus shared features across train and validation set can result in inflated validation set performance in model evaluation. We carefully read through the articles provided with the comment, and found that both of them were concerned with the incorrect report of model performance due to the shared enhancer features across the train and validation set.

In fact, we were already aware of this issue of information leak and made the evaluation of the Chromoformer model carefully designed to avoid the problem. We tried to prevent it by splitting genes into train and validation sets according to the chromosome in which each gene is located so that no cis-regulatory elements are shared across genes in train and validation sets. Please note that we do not consider any trans-interactions between regulatory elements to ensure that no information is allowed to be transferred across different chromosomes.

Minor comments

1. Figure 2 – “CV” is not explained in the figure legend

Following this comment, we added a description in the legend of Figure 2 in the manuscript.

2. I am still not very clear about how the training dataset was prepared. The authors say “For each cell type, the median expression values across all genes were used as threshold values to assign genes with one of the two labels: highly (1) or lowly expressed (0).” – does this mean that if the gene expression was above the median expression, then it was denoted as “1”?

According to this comment, we elaborated the label assignment procedures in the Methods section of the revised manuscript. As the reviewer has mentioned, if a gene had expression above median expression in that cell type, it was assigned with label “1”, otherwise it was assigned with label “0”. Importantly, these labels “1” and “0” do not have any quantitative meanings (i.e., “1” does not inform the model that the gene is expressed as an amount of “1”). Instead, they just denote the ordinal indices of binary classification labels. In other words, “0” indicates that the gene is assigned to the first class, and “1” indicates that the gene is assigned to the second class. Therefore, we can expect that a Chromoformer model, which is trained to discriminate between the two distinct classes, will be trained exactly the same even if we swap label assignment of genes (“0” to “1” and “1” to “0”), or use labels “-1” and “1” instead of “0” and “1”.

Reviewer #3 (Remarks to the Author):

In this article, the authors present a transformer-based, three-dimensional (3D) chromatin conformation-aware deep learning architecture named Chromoformer that achieves the state-of-the-art performance in the quantitative deciphering of the histone codes in gene regulation. The authors decomposed the architecture into three phrases understand the complex dynamics of cis-regulations involving multiple layers (1) cis-regulation by core promoters, (2) 3D pairwise interaction between a core promoter and a putative cis-regulatory regions (pCREs) and (3) a collective regulatory effect

imposed by the set of 3D pairwise interactions.

Here are my reviews for this article

1. The authors developed three transformer models (Embedding transformer, pairwise transformer and Regulation Transformer). For each transformer model, the input features are core promoter features and pCRE features. It is unclear how these features were generated. Did any deep learning models were used to generate these features? Some details are necessary to understand this step.

We appreciate this comment for pointing out unclear methodological descriptions in the manuscript. According to the reviewer's suggestion, we elaborated the feature generation procedure in Supplementary Figure 1 of the revised manuscript. The input feature generation process does not require any deep learning models, but only utilizes a common bioinformatics pipeline for next generation sequencing read alignment and genomewide read depth calculation.

Here, we illustrate the feature generation procedure in detail (**Figure R3-1**). As the reviewer has mentioned in the comment, the only input features required for the Chromoformer model are the abundances of seven major HMs (H3K4me1, H3K4me3, H3K9me3, H3K27me3, H3K36me3, H3K27ac, H3K9ac) within promoters as well as cis-regulatory elements associated with them. Most importantly, the abundance of a certain HM was first computed for each base pair throughout the genome as the number of histone ChIP-seq reads (or read depth) covering that position (**Figure R3-1a**). Indeed, these 1bp-resolution HM signals can be used for input features. However, since the memory requirement of the self-attention operation in Chromoformer architecture scales quadratically, it is not feasible to use 40,000bp-long sequences directly as input features. Therefore, we instead used bin-level average histone modification signals so that we can deal with shorter sequences (sequence with 400, 80, 20 entries for 100bp, 500bp, 2,000bp bins, respectively). For core promoter features, bin-averaging the histone signals for the 40kbp-region centered at TSS yields (7 x 400), (7 x 80) and (7 x 20) matrices for 100bp, 500bp and 2000bp resolutions, respectively (**Figure R3-1b**). Similarly, for each pCRE feature, we also have (7 x 400), (7 x 80) and (7 x 20) matrices (**Figure R3-1c**). However, since the length of a pCRE is determined by the length of HindIII restriction fragments, zero-padding is needed on the left and right side of the matrices to have constant size of input matrices.

2. The architecture of "Embedding Transformer" and "Pairwise Interaction Transformer" is same. Is there any specific reason to give different name? It could be misleading that you have developed two different transformer architecture.

We thank the reviewer for raising important concerns about the nomenclature of model substructures. The reason why we decided to give those two substructures (Embedding and Pairwise Interaction transformers) separate names can be explained by the following two aspects of differences between them: (1) Difference in the type of attention operation used, and (2) difference in learning semantics.

1. Difference in the type of attention operation

Figure R3-1 (Related to Supplementary Fig. 1). Input feature generation procedures. (a) Preparation of histone modification signals. (b) Generation of core promoter features. (c) Generation of core pCRE features.

The critical difference between Embedding and Pairwise Interaction transformers is that the former is essentially based on self-attention operation, and the latter is based on encoder-decoder attention. Here, we explain the differences between those two variants of attention operations to emphasize the difference between Embedding and Pairwise Interaction transformers.

The core operation for all the three types of transformers in Chromoformer is the Query-Key-Value attention (denoted as red boxes labeled with “Multi-Head Attention” in Figure 1c-e in the manuscript). Briefly, Query-Key-Value attention produces the updated version of query embeddings as the weighted sum of Value embeddings. Here, the weights are determined through the computation of affinities between Query and Key embeddings. The critical difference between self-attention and encoder-decoder attention is that self-attention generates both Query

and Key embeddings from a single sequence (or set of vectors), while encoder-decoder attention generates Query and Key embeddings separately from two different sequences. Therefore, self-attention measures the “affinities” between two positions within a single sequence, while encoder-decoder attention measures the affinities between two positions at two independent sequences. This apparently small difference results in a crucial difference in the semantics of Chromoformer learning, which is discussed in the following.

2. Semantic difference.

Since the core operation within the Embedding transformer and Pairwise Interaction transformer is different, what they are designed to learn is also different. An Embedding transformer only takes a core promoter feature as an input, and is trained to capture the intra-dependencies of HM configurations at different positions within the given core promoter. On the other hand, a Pairwise Interaction transformer takes a pair of a core promoter and a corresponding pCRE as input, and learns the pairwise dependencies between the two positions in the core promoter and the pCRE.

Nevertheless, we agree that the names we proposed might mislead the readers to expect clear architectural differences between the Embedding and Pairwise transformers. Therefore, we clarified the similarities and differences between the three types of transformers in Chromoformer (Embedding, Pairwise Interaction and Regulation transformers) in the Supplementary Information of the revised manuscript.

3. One of the main key points of the regulation transformer is to “normalized interaction frequencies f between the corresponding core promoter-pCRE pair as a bias term to the self-attention matrix to inform the model with the relative affinities of the pairwise interactions”. What is the reason of using “normalized interaction frequencies”? How were normalized interaction frequencies derived?

Given the reviewer’s comment, we noticed that the rationale and description for the normalization of interaction frequencies was not clear enough in the manuscript.

We decided to use normalized interaction frequencies instead of raw interaction frequencies because there are some technical biases in raw interaction frequencies that hampers the direct interpretation of those values. First, due to the regional preference of a sequencing experiment, restriction and alignment methods, the coverage or mappability of Hi-C sequencing reads throughout the genome is not uniform. This is exacerbated in pcHi-C experiments since the fragment containing the promoter is significantly high due to promoter-enrichment procedure (For example, the raw coverage of promoter fragment is about 14.4 times higher than non-promoter fragments for H1 pcHi-C data). Thus, the frequencies of promoter-promoter interactions would be more exaggerated than the true amount of interactions between them. Next, the random Brownian motion of DNA polymer results in higher frequency of non-biological interactions between the two fragments at closer linear distance along the genome. This distance bias should be corrected because otherwise the results would erroneously favor interactions at close distances and ignore long-range biological contacts such as promoter-enhancer interactions.

Regarding the two aforementioned biases, normalized interaction frequencies were obtained by statistically correcting them. First, the coverage bias is corrected by fitting a negative binomial regression model for raw ligation frequencies between two fragments using individual coverage values. Formally, the raw interaction frequency (i.e., read ligation frequencies) between two DNA fragments i and j , Y_{ij} , is normalized using the coverages C_i and C_j as follows. Using values of Y_{ij} ,

Figure R3-4 (Related to Supplementary Fig. 4). Chromoformer-clf model performances when self-attention-based aggregation of regulatory embeddings was used instead of concatenation. (a) Schematic illustration of self-attention operation proposed by Lin et al. **(b)** Schematic illustration of scaled dot-product attention proposed by Vaswani et al. **(c)** Performances of Chromoformer-clf models.

the expected interaction frequency u_{ij} is fitted by negative binomial regression model $\log(u_{ij}) = \beta_0 + \beta_1 C_i + \beta_2 C_j$. Then, the normalized interaction frequency R_{ij} is obtained by taking residual $R_{ij} = \frac{Y_{ij}}{\exp(\beta_0 + \beta_1 C_i + \beta_2 C_j)}$.

Subsequently, distance bias is corrected in a similar manner. Given the linear distance between two genomic fragments i and j , D_{ij} , the expected ligation frequency was fitted by negative binomial regression model $\log u_{ij} = \beta_0 + \beta_1 D_{ij}$. When $D_{ij} = d$, the expected ligation frequency is given by $E_d = \exp(\beta_0 + \beta_1 d)$. Therefore, the distance-dependent signal can be removed by taking residual $(R_{ij} + \text{avg}(R_{ij})) / (E_d + \text{avg}(R_{ij}))$, where $\text{avg}(R_{ij})$ is a global average value of R_{ij} 's.

We added the description for the normalization of interaction frequencies in the Supplementary Information of the revised manuscript. Of note, the normalization procedure above is implemented in the R package `covNorm` (Kim et al., 2021).

4. The authors used three modules (100bp, 500bp and 2000bp modules) and the outputs concatenated before final prediction using fully connected layers. Can self-attention based concatenation be applied so that the multi-scale regulatory embedding is generated with their attention score? As in the normal concatenation, are you giving equal weightage to all three outputs?

We appreciate this insightful suggestion on the model architecture. We agree that there may exist meaningful interactions between different scales (or resolutions) of the combinations of histone

modifications that contribute to gene regulation. We also thought that these interactions, if they exist, can be effectively captured by an attention operation between regulatory embeddings representing multiple scales. Therefore, we measured the model performances after substituting the concatenation of regulatory embeddings with two different self-attention mechanisms according to the reviewer's suggestion.

First, we adopted the self-attention mechanism proposed in Lin et al. (2017), whose formulation is similar to "additive" attention suggested by Luong et al. and Bahdanau et al. (**Figure R3-4a**). Of note, this attention operation is used in the benchmark model HM-CRNN (Kang et al. (2020)) in order to attend to relevant genomic positions in the gene expression prediction task. More formally, if we let 256-dimensional regulatory embeddings of 100bp, 500bp and 2000bp resolution (i.e., outputs of 100bp-, 500bp- and 2000bp-resolution modules) be x_1 , x_2 and x_3 , respectively, the attention operation to combine the three embeddings is given as follows. First, the embeddings are first projected to produce attentional hidden states h_i .

$$h_i = \tanh(W_1 x_i), \quad i = 1, 2, 3$$

where W_1 denote a weight matrix of size $256 \times t$, respectively. In this experiment, t is set to 16. Then attention score for i -th resolution, α_i , is computed as below:

$$\alpha_i = \text{Softmax}\left(\frac{W_2 h_i}{\sum_{i=1}^3 W_2 h_i}\right)$$

After computing α_1, α_2 and α_3 , the final embedding is obtained as a weighted sum:

$$x = \sum_{i=1}^3 \alpha_i x_i$$

Finally, x is fed to the following full-connected head to produce binary logits as in the original Chromoformer model.

Next, we also examined the effect of another form of self-attention, namely scaled dot-product self-attention (Vaswani et al., 2017), which has already been used at the core of transformer modules in Chromoformer (**Figure R3-4b**).

As a result, we did not observe significant performance improvement of attention-based aggregation over concatenation-based aggregation of three regulatory embeddings (**Figure R3-4c**). Based on this result, we could deduce that the fully connected layers downstream the concatenation operation have sufficient capacity to learn the nonlinear interactions between latent features of different resolutions. We added this discussion to the revised manuscript (Supplementary Fig. 4).

5. Also, why the authors specifically consider 100bp, 500bp and 2000bp input regions? Did the authors check the prediction result only for two regions like 100bp and 500bp OR 500bp and 2000bp?

We appreciate the reviewer's suggestion. According to the comment, we conducted additional experiments by training Chromoformer models using every combination of input resolutions (i.e., using only a single resolution and also using a combination of two out of three input resolutions) to justify our choice of using all the three input resolutions. We added the results in the revised manuscript (Supplementary Figure 3).

Figure R3-5 (Related to Supplementary Fig. 3). Chromoformer model performances when different combinations of resolutions were used.

As a result, we observed that the models using all the three resolutions altogether (2000bp, 5000bp and 100bp) were among the top 2 performing models in every cell type examined (**Figure R3-5**). We note that the combination of (2000bp and 500bp) or (2000bp and 100bp) resulted in comparable performances in some cases, but we considered it best to use the combination of all the three resolutions because it showed robustly high performances across cell types.

References

Avsec, Ž., Agarwal, V., Visentin, D., Ledsam, J. R., Grabska-Barwinska, A., Taylor, K. R., ... & Kelley, D. R. (2021). Effective gene expression prediction from sequence by integrating long-range

interactions. *Nature methods*, 18(10), 1196-1203.

Bahdanau, D., Cho, K., & Bengio, Y. (2014). Neural machine translation by jointly learning to align and translate. *arXiv preprint arXiv:1409.0473*.

Chan, H. L., & Morey, L. (2019). Emerging roles for polycomb-group proteins in stem cells and cancer. *Trends in biochemical sciences*, 44(8), 688-700.

Dixon, J. R., Selvaraj, S., Yue, F., Kim, A., Li, Y., Shen, Y., ... & Ren, B. (2012). Topological domains in mammalian genomes identified by analysis of chromatin interactions. *Nature*, 485(7398), 376-380.

Kahn, T. G., Dorafshan, E., Schultheis, D., Zare, A., Stenberg, P., Reim, I., ... & Schwartz, Y. B. (2016). Interdependence of PRC1 and PRC2 for recruitment to Polycomb Response Elements. *Nucleic acids research*, 44(21), 10132-10149.

Kim, K., & Jung, I. (2021). covNorm: An R package for coverage based normalization of Hi-C and capture Hi-C data. *Computational and Structural Biotechnology Journal*, 19, 3149-3159.

Kubo, N., Ishii, H., Xiong, X., Bianco, S., Meitinger, F., Hu, R., ... & Ren, B. (2021). Promoter-proximal CTCF binding promotes distal enhancer-dependent gene activation. *Nature structural & molecular biology*, 28(2), 152-161.

Kundaje, Anshul, et al. "Integrative analysis of 111 reference human epigenomes." *Nature* 518.7539 (2015): 317-330.

Lin, Z., Feng, M., Santos, C. N. D., Yu, M., Xiang, B., Zhou, B., & Bengio, Y. (2017). A structured self-attentive sentence embedding. *arXiv preprint arXiv:1703.03130*.

Luong, M. T., Pham, H., & Manning, C. D. (2015). Effective approaches to attention-based neural machine translation. *arXiv preprint arXiv:1508.04025*.

Schmitt, A. D., Hu, M., Jung, I., Xu, Z., Qiu, Y., Tan, C. L., ... & Ren, B. (2016). A compendium of chromatin contact maps reveals spatially active regions in the human genome. *Cell reports*, 17(8), 2042-2059.

Sekhon, A., Singh, R., & Qi, Y. (2018). DeepDiff: DEEP-learning for predicting DIFFerential gene expression from histone modifications. *Bioinformatics*, 34(17), i891-i900.

Singh, S., Yang, Y., Póczos, B., & Ma, J. (2019). Predicting enhancer-promoter interaction from genomic sequence with deep neural networks. *Quantitative Biology*, 7(2), 122-137.

Vaswani, A., Shazeer, N., Parmar, N., Uszkoreit, J., Jones, L., Gomez, A. N., ... & Polosukhin, I. (2017). Attention is all you need. *Advances in neural information processing systems*, 30.

Reviewers' Comments:

Reviewer #1:

Remarks to the Author:

The authors addressed all my comments and clarify the performance of the new method. I think the revised manuscript is highly improved and provides new insight. One of the interesting points that should be discussed is the marginal combined effect of H3K27ac and H3K4me1 (Supple Fig. 10). As these histone modifications are key marks for enhancer elements, it would be valuable to discuss how enhancer markers are not critical to predicting gene expression patterns.

For other comments, several supplementary figures related to the cross-species and cross-cell type analyses would be worth being included in the main figures.

I appreciate the authors for their valuable work in this research area.

Reviewer #2:

Remarks to the Author:

My concerns have been addressed.

Reviewer #4:

Remarks to the Author:

I thank the authors to provide an elaborate and clear explanation on all the questions asked.

please note the following typo's:

line 239: ...by the its...

line 364: ...exists a certain ...

line 518: ...from from...

Response to reviewer's comments

We would appreciate all the reviewers for their constructive comments, which motivated us to conduct further analyses and helped us to improve our manuscript. In this revision, we added a discussion on multiple feature ablation experiment and also added Figure 8 to help readers with cross-species and cross-cell type performance of Chromoformer in the Discussion section. Please see below our detailed response to reviewers' comments in blue.

Reviewer #1 (Remarks to the Author):

The authors addressed all my comments and clarify the performance of the new method. I think the revised manuscript is highly improved and provides new insight. One of the interesting points that should be discussed is the marginal combined effect of H3K27ac and H3K4me1 (Supple Fig. 10). As these histone modifications are key marks for enhancer elements, it would be valuable to discuss how enhancer markers are not critical to predicting gene expression patterns.

We appreciate the insightful comment. We were also curious why ablating well-known enhancer marks H3K27ac and H3K4me1 at the same time did not result in significant performance degradation. Therefore, we further scrutinized the spatial correlation between HMs (**Supplementary Figure 10b**) and the patterns of multiple-feature ablation experiment (**Supplementary Figure 10c**). We noticed the co-occurrence of active HMs (H3K4me1, H3K4me3, H3K27ac and H3K9ac; **Supplementary Figure 10b**) and considerable performance drop when those four HMs are simultaneously removed from model training (**Supplementary Figure 10c**, sixth row). Notably, the amount of validation AUC decrease was even greater than many of H3K36me3-ablated cases. However, when any of the four active marks are once included in the training, the performance degradation seemed to be greatly alleviated. Altogether, the redundancy of active HMs in part explains why the ablation of H3K27ac and H3K4me1 is not solely sufficient to degrade the performance of Chromoformer. We especially conjecture that the promoter-promoter (P-P) interactions between active promoters marked by H3K4me3 or H3K9ac may be hinting the existence of transcription factories enriched with enhancers, compensating the absence of H3K27ac and H3K4me1 marks. Nevertheless, we note that ablating both H3K27ac and H3K4me1 with additional HMs showed relatively high performance drop (without H3K36me3 ablation) and added indications for that in **Supplementary Figure 10c** (green box). Following the reviewer's suggestion, we added additional discussion to a subsection named "Chromoformer learns to attend to the distant transcriptional elongation signals at gene bodies" regarding Supplementary Figure 10.

For other comments, several supplementary figures related to the cross-species and cross-cell type analyses would be worth being included in the main figures.

According to the reviewer's suggestion, we added an additional Figure (**Figure 8**) including cross-species and cross-cell type analyses results to improve the readability of the Discussion section of the manuscript.

I appreciate the authors for their valuable work in this research area.

We thank the reviewer for the considerate comments.

Reviewer #2 (Remarks to the Author):

My concerns have been addressed.

Reviewer #4 (Remarks to the Author):

I thank the authors to provide an elaborate and clear explanation on all the questions asked.

please note the following typo's:

line 239: ...by the its...

line 364: ...exists a certain ...

line 518: ...from from...

We fixed the typos accordingly.